# Reversible morphology-resolved chemotactic actuation and motion of Janus emulsion droplets

Bradley D. Frank [1], Saveh Djalali [1], Agata W. Baryzewska[1], Paolo Giusto [1], Peter H. Seeberger[2] & Lukas Zeininger [1]✉

We report, for the first time, a chemotactic motion of emulsion droplets that can be controllably and reversibly altered. Our approach is based on using biphasic Janus emulsion droplets, where each phase responds differently to chemically induced interfacial tension gradients. By permanently breaking the symmetry of the droplets' geometry and composition, externally evoked gradients in surfactant concentration or effectiveness induce anisotropic Marangoni-type fluid flows adjacent to each of the two different exposed interfaces. Regulation of the competitive fluid convections then enables a controllable alteration of the speed and the direction of the droplets' chemotactic motion. Our findings provide insight into how compositional anisotropy can affect the chemotactic behavior of purely liquid-based microswimmers. This has implications for the design of smart and adaptive soft microrobots that can autonomously regulate their response to changes in their chemical environment by chemotactically moving towards or away from a certain target, such as a bacterium.

[1] Department of Colloid Chemistry, Max Planck Institute of Colloids and Interfaces, Am Muehlenberg 1, 14476 Potsdam, Germany. [2] Department of Biomolecular Systems, Max Planck Institute of Colloids and Interfaces, Am Muehlenberg 1, 14476 Potsdam, Germany. ✉email: lukas.zeininger@mpikg.mpg.de

Autonomous regulation of chemotactic motility represents a fundamental ability of living organisms. Cells, for instance, are capable of interacting with and responding to small changes in their chemical environment by translating specific chemical recognition events at their surface into an oriented chemotactic motion. Striking examples of this autonomous behavior include white blood cells that use their unique chemotactic ability to search, sense, and react to external insults and thereby fundamentally contribute to our body's inflammatory response. Directional chemotactic locomotion represents a key feature of inter-colloidal communication and cooperation, and as such provides a fundamental mechanism for Nature's ability to design adaptive chemo-mechanical feedback networks[1].

The design of artificial microswimmers that can undergo an externally triggered or spontaneously induced chemotactic motion targets the design of bodies that can emulate the autonomous behavior observed within natural systems[2,3]. Such bodies can act as artificial model systems to better understand the fundamental mechanisms of inter-colloidal communication within biological systems. Artificial microswimmers have further garnered considerable interest for potential applications in drug and cargo delivery, biosensing, environmental remediation, precise manipulation of objects, including in medical surgery or manufacturing, as well as to study complex emergent properties arising from the interaction of stimuli-responsive multibody systems such as swarm behavior[4–9].

Many of these applications rely on the ability to control, program, configure, and direct the motion of microswimmers as opposed to exhibiting random Brownian motion[10]. Central to the realization of stimuli-directed motility is the evocation of an out-of-equilibrium state within the colloidal body, such as via chemical concentration gradients or physical triggering events. Utilizing these dynamics, a series of artificial soft-matter robots on the micro- or nanoscale powered by chemical fuels have been reported. Motion can be mediated by (photo-)catalytic reactions at their interface[11,12], surface-encoded translation pathways[13], by bacteria[14], or via different physical stimuli, including light, sound, magnetic, or electric fields[15–19].

Emulsion droplets are appealing models as they intrinsically represent a thermodynamically metastable system. Emulsions are highly dynamic systems, where the constant exchange of molecules with their environment provides an opportunity for communication and reactivity between the object and the environment[20–24]. In addition, the suitability of droplets as biomimetic model systems is underpinned by their similarity in size, and the organization of amphiphiles to the droplet interfaces, which resembles the environment found at cell surfaces[25,26].

Droplets can chemotax in response to small interfacial tension variations, driven by Marangoni-type fluid flows[27–29]. Imbalances in interfacial tension across the droplet surface cause Marangoni-type fluid convections from regions of low interfacial energy to regions of high interfacial energy. These fluid flows develop along the droplet interface, which induce a commensurate convective flow inside the droplets. Small lateral variations in interfacial tension thereby suffice to achieve significant Marangoni stress-induced fluid currents at the interface that can propel a motion of the droplets[30]. The induced currents occur in the direction of regions with higher surface tension, which drives a chemotactic locomotion of the emulsion droplets in the opposite direction. Interfacial tension gradients across a droplet surface can be evoked either chemically by variation of the surfactant effectiveness or concentration[31–35], or via fluctuations in the surfactant density at the droplet interface caused by anisotropic micellar solubilization of the dispersed phase[36–39]. Depending on the underlying propulsion mechanism, the directionality of the overall droplet movement can be random or unidirectional[20,36,39–43], with the

motion being sustained until the transient interfacial tension anisotropy equilibrates, at which point the droplets become stationary.

An anisotropic dispersed phase solubilization into surfactant micelles can cause gradients in the surfactant density across a droplet interface, which initiate Marangoni-type convective fluid flows that drive droplet motion. In this mechanism, droplet motion can be sustained and droplets become self-propelled due to repulsion away from their own swollen micelle trail. Droplet velocities up to 400 μm s$^{-1}$ have been reported for this propulsion mechanism[44]. In turn, for surfactant-stabilized droplets responding to an external chemical gradient, velocities up to 21,000 μm s$^{-1}$ have been reported[27,45,46]. In this mechanism, droplets act as motile sensors to interfacial tension gradients caused by local changes in the surfactant composition, strength, or effectiveness, for instance, caused by chemical reactions of stimuli-responsive surfactants. Marangoni-type fluid flows from regions of low interfacial energy to regions of high interfacial energy drive the droplet locomotion. Consequently, single-phase droplets of isotropic composition display selective unidirectional motion in the direction towards a minimization of interfacial energy and such approaches have yet to create emulsion droplets that can reversibly and controllably alter speed and direction of motion.

We hypothesized that by introducing anisotropy to the internal droplet morphology and composition, an asymmetry in the fluid flows surrounding a droplet and thus a controllable selective and dynamically tunable directionality in the locomotion of emulsion droplets can be achieved. Whereas solid Janus colloids with two different chemistries on each side of the particle have found widespread application in the design of active microswimmers with directional propulsion profiles[47], a permanently asymmetric, dynamic, and fluid-based system such as spherical biphasic Janus droplets has not been explored previously in this context. Our Janus emulsion droplets are comprised of two phase-separated oils that align with gravity, dispersed within an aqueous continuous phase. We demonstrate that due to the anisotropy of the internal droplet composition, externally induced or (bio-)chemically evoked surfactant concentration gradients cause differing interfacial tension gradients on each hemisphere of the droplets. As a result, depending on the chemical stimuli and the nature of the surfactants, arising Marangoni-type fluid flows can be unidirectional or competitive, i.e. pointing into the opposite direction. Precise manipulation of the fluid flows, mutually controlled by the differing interfacial tension gradients as well as the exposed surface areas of the individual droplet phases, then allows to controllably, and reversibly direct the droplets' chemotactic motion up or down a chemical concentration gradient, or synonymously towards or away from a (bio-)chemical source. The ability to design reconfigurable liquid systems capable to chemotactically respond to chemical gradients reversibly, provides a novel strategy for the design of autonomously operating, adaptive, and chemo-intelligent soft materials.

## Results

**Droplet chemotaxis in binary surfactant gradients**. To investigate the chemotactic motion of emulsion droplets in response to an externally induced interfacial tension gradient, we place oil droplets dispersed within a surfactant-containing aqueous continuous phase (~ 150 μL) inside a sealable milled polycarbonate/glass microfluidic chip composed of two wells with a microchannel ($d = 300$ μm; $L = 20$ mm) between them, as schematically displayed in Fig. 1a and S1. The sealed channel allows for 'free' movement of the droplets on the flat bottom glass substrate, and circumvents evaporative volume changes and capillary flows.

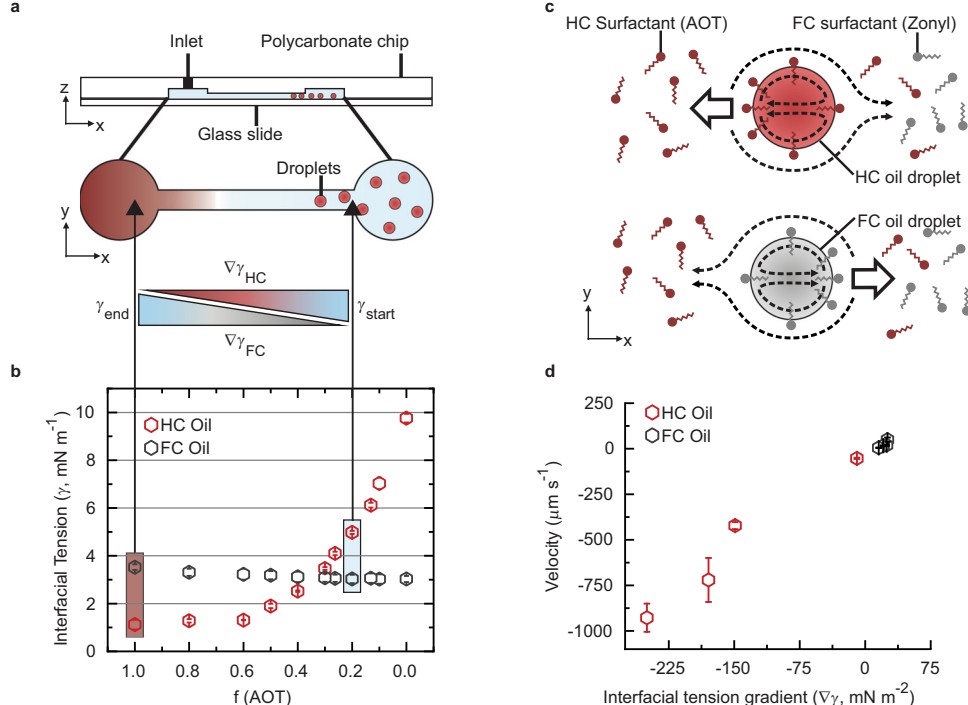

**Fig. 1 Chemotactic motion of single-phase emulsion droplets within a binary surfactant gradient of constant overall concentration. a** Schematic of the microfluidic channel used in these experiments, where droplets placed inside the channel experience a laminar interfacial tension gradient upon addition of a surfactant solution with different composition to the open end of the channel; **b** interfacial tensions of the two different HC and FC droplet oils (HC: 1:3 mixture of decane and bromohexane; FC: methoxyperfluorobutane) inside an overall 1 wt.% binary surfactant solution of AOT and Zonyl, displayed as a function of the fraction of AOT, $f$ (AOT), where the other fraction is Zonyl; **c** schematic diagram of Marangoni-type fluid flow fields (dashed lines) induced by a lateral interfacial tension gradient at the droplet interface and the resulting direction of chemotactic motion of the droplets with respect to the surrounding liquid (large arrow). The different surfactant densities at the droplet interface are used to schematically represent the induced interfacial tension gradient at the droplet interface.; **d** experimentally determined velocities of droplets stabilized by 1 wt.% solutions of AOT and Zonyl in different compositions, upon addition of a pure 1 wt.% AOT solution to the open end of the microchannel, plotted against the estimated interfacial tension gradient from interfacial tension measurements; a negative velocity reflects a droplet motion towards the open end of the channel and a positive velocity corresponds to a movement away from the added surfactant solution; error bars reflect > 5 measurements.

To evoke a directional surfactant gradient inside the microchannel, we add an additional 10 μL of a solution comprising a different surfactant concentration or mixture to the open end of the channel at the opposing side of the droplets. Diffusion of the surfactant solution through the microchannel, observed on the order of 2–3 min, then creates a laminar surfactant gradient (Supplementary Fig. 13, Supplementary Movie 10).

A chemotactic response of single phase emulsion droplets to a pure surfactant concentration gradient has been previously observed and studied both theoretically and experimentally,[27,42,43,45,46] We reasoned that a similar chemotactic motility due to an induced laminar interfacial tension gradient can also be evoked inside a binary surfactant solution where the ratio of two different surfactants is altered, whereas the overall surfactant concentration remains constant. To this end, we employ microfluidics to generate monodisperse single phase droplets with an average size of $d = 102 \pm 8$ μm, comprised of a homogeneous mixture of decane and 1-bromohexane (1:3) dispersed within an aqueous continuous phase containing 1 wt.% of the two surfactants sodium dioctyl sulfosuccinate (AOT) and Zonyl FS-300 (Zonyl) in a 2.5:7.5 ratio. The specific oil mixture ensures a density of the dispersed phase slightly larger than that of the continuous phase.

Next, to initiate chemotactic motion, a more concentrated AOT solution (1 wt. %) is added to the inlet of the channel. In these experiments, we observe a rapid movement of the droplets inside the channel towards the region of the added surfactant solution. To analyze the stimulated chemotactic motion of the

droplets, a video of the moving droplets is captured via an optical microscope over a uniform section of the channel between the two wells. Droplet trajectories and velocities are then analyzed using a modified MATLAB-based tracking software, where droplet velocities are extracted from freely moving, singular droplets after initial acceleration from still (Supplementary Fig. 2)[48]. In agreement with theoretical considerations, which predict high potential droplet velocities of several micrometers per second with only small interfacial tension gradients (Supplementary Fig. 3), we observe velocities of 720 μm s$^{-1}$ in these original tests (Fig. 1d).

To explain the rapid droplet motion in response to an induced interfacial tension gradient, we determined the interfacial tensions of the respective oil mixture inside the original 2.5:7.5 AOT:Zonyl solution ($\gamma = 4.71$ mN m$^{-1}$) and inside a pure 1 wt.% AOT solution ($\gamma = 1.12$ mN m$^{-1}$) using pendant drop tensiometry (Fig. 1b). The strong measured contrast for the two boundary cases and the significantly lowered interfacial tension inside the added solution serves to explain the rapid droplet motion toward the open end of the channel. Interfacial tension differentials across a droplet interface evoke convective Marangoni-type fluid flows from regions of low interfacial energy to regions of high interfacial energy that drive the droplet locomotion (Fig. 1c). Alterations of the dynamic interfacial tension gradient inside the microchannel via changing the surfactant concentration of the original or added surfactant solution reveal the previously reported linear proportionality between $\nabla\gamma$ and droplet velocity (Fig. 1d).

In addition, using the same experimental setup, we tested a second droplet system consisting of the fluorocarbon oil methoxyperfluorobutane (MPFB). In contrast to decane:bromohexane droplets that selectively moved towards regions of higher hydrocarbon surfactant ratio, in these experiments, we observe droplet motion in the opposite direction. The chemotactic motion of the fluorocarbon oil droplets away from the added hydrocarbon surfactant is attributed to the driving force of lowering their interfacial tension. Although the measured interfacial tensions for the MPFB oil inside pure AOT and Zonyl solutions displayed only small differences of $\gamma = 3.03 \text{ mN m}^{-1}$ and $\gamma = 3.53 \text{ mN m}^{-1}$, respectively, this interfacial tension gradient is sufficient to induce droplet motion away from the added hydrocarbon surfactant. A comparison of the resulting droplet velocities reveals overall slower velocities of the fluorocarbon droplets due to the marginal gradients in interfacial tension (Fig. 1d).

**Directional chemotaxis of Janus emulsion droplets**. With a basic understanding of droplet motility within binary surfactant mixtures at hand, we next set out to apply the same experimental conditions to complex Janus emulsion droplets. To this end, we generate monodisperse biphasic Janus emulsions comprised of a phase-separated 1:1 mixture of the two oils decane (HC) and methoxyperfluorobutane (FC) inside a microfluidic channel using a thermal phase separation approach (Fig. 2a and Supplementary Fig. 4)[49]. The particular fluid combination was selected as it presents a large density contrast that ensures a gravitational alignment of the droplet phases, placing the denser FC phase at the bottom. The overall droplet density of the mixture however is slightly higher than that of water, allowing droplets to sink inside an aqueous continuous phase but guarantees free movement of the droplets on the glass surface. For these particular emulsions, the overall droplet shape can be assumed to be spherical, as the interfacial tension between the droplet phases is much smaller than the interfacial tensions between the droplet constituent oil phases and the aqueous continuous medium[50,51]. The internal Janus droplet morphology is determined by the force balance of interfacial tensions acting at the individual interfaces and changes in the surfactant composition does not affect the overall spherical droplet stability, but allow for a controllable modulation and alteration of the internal droplet geometry[52,53].

When placing the Janus emulsion droplets inside the laminar surfactant gradient within the microchannel, depending on the initial starting morphology of the droplets we observe a vastly differing behavior. Whereas droplets with an 'opened-up' Janus morphology (triple phase contact angle $\theta = 150°$; for calculation of the contact angle see Supplementary Fig. 4) move towards the open end of the channel where we added a 1 wt.% pure AOT solution (Fig. 2b, Supplementary Movie 1), droplets of inverse morphology ($\theta = 30°$) move away from the added surfactant solution (Fig. 2c and Supplementary Movie 2). In both cases, droplet acceleration starts after a few seconds and this initial time lapse is attributed to diffusion of the added surfactant into the channel.

To understand the differences in the directionality of droplet motion we start by comparing the interfacial tensions of both individual droplet phases inside 1 wt.% surfactant solutions of different compositions (Fig. 2d). The measurements reveal a strong dependency of the interfacial tension of decane on the respective surfactant composition with significantly lower values measured inside a pure AOT-containing aqueous phase. This difference is less pronounced for methoxyperfluorobutane however, with slightly lower interfacial tension values inside Zonyl-containing continuous phases. These measurements illustrate that the two external interfaces of Janus droplets experience opposite

interfacial tension gradients ($\nabla \gamma$) when placed inside a laminar gradient in surfactant composition inside the channel. We conclude that the resulting Marangoni-type fluid flows on the upper HC hemisphere and the bottom FC hemisphere of Janus droplets are directed into the opposite direction and thus in competition to each other. The overall droplet movement and its direction is then determined by the dominant flux of the two induced competitive fluid flows.

To visualize the competitive fluid flows surrounding a Janus droplet exposed to a surfactant gradient, we disperse tracer particles into the surrounding aqueous continuous phase and map the flow fields using particle imaging velocimetry. When subjected to a surfactant gradient, substantial fluid motion is perceived in the vicinity of the biphasic droplets (Fig. 2e). The flow profiles reveal that the direction of the Marangoni flows adjacent to each of the two different phases of a Janus droplet point in the opposite direction for each phase, from the region of low to regions of high interfacial tension. In these experiments, tracer particles invert their movement direction at the triple phase contact line, independent of the droplet internal morphology (Supplementary Fig. 5). The two competitive fluid flows adjacent to each phase result in droplet displacement forces pointing in opposite directions and we concluded that, at low Reynolds number, the directionality and the speed of the overall droplet displacement overcoming Stokes drag is dictated by the net force resulting from the two competitive Marangoni flows.

To reveal the dependency of directionality and velocity of the droplet motion solely on the power of the individual anisotropic Marangoni flows, we record experimentally the Janus droplet motion as a function of different surfactant compositions in the originally placed continuous phase and thus varying induced interfacial tension gradients and starting droplet morphologies (Supplementary Fig. 6). As the graph reveals (Fig. 2g), when we add a pure AOT solution to initiate movement, droplets in a perfect Janus morphology ($\theta = 90°$, original surfactant composition: $f(AOT) = 0.36$) move towards the open end of the channel, as a result of a dominant flux adjacent to the HC hemisphere. Comparing determined droplet velocities with previous measurements for single phase HC droplets reveal that the overall Janus droplet motion is significantly slower, which corroborates the presence of a competitive force originating from the Marangoni flow adjacent to the FC phase.

Upon expansion of the HC-W interface of droplets placed inside the channel via increasing the AOT fraction of the originally placed continuous phase prior to initiating a surfactant gradient, we observe an increase in the measured droplet velocities. This is despite the fact that the interfacial tension difference of the HC phase between the original and added surfactant solution is smaller, which would consequently result in a lowered displacement force. These experiments illustrate that the force balance between the competitive fluid flows determining the droplet chemotactic motion is not only dependent on the respective interfacial tension gradients at the two interfaces but also the ratio of exposed surface areas. Assuming a linear local interfacial tension gradient across a droplet interface, which is an appropriate approximation given that the diameter of the droplets are much smaller than the overall channel length, the net force acting on a droplet due to the competitive interfacial tension gradient-induced Marangoni flows can be expressed as

$$m\frac{\mathrm{d}v_{\text{drop}}}{\mathrm{d}t} = F_{\text{HC}} + F_{\text{FC}} - F_{\text{D}} \cong \int_A \frac{\delta\gamma_{\text{HC}}}{\delta x} + \int_A \frac{\delta\gamma_{\text{FC}}}{\delta x} - 6\pi r\eta v_{\text{drop}}$$

(1)

where m is the droplet mass, $v_{\text{drop}}$ is the droplet velocity, $F_{\text{HC}}$ and $F_{\text{FC}}$ are the forces adjacent to the individual Janus droplet phases,

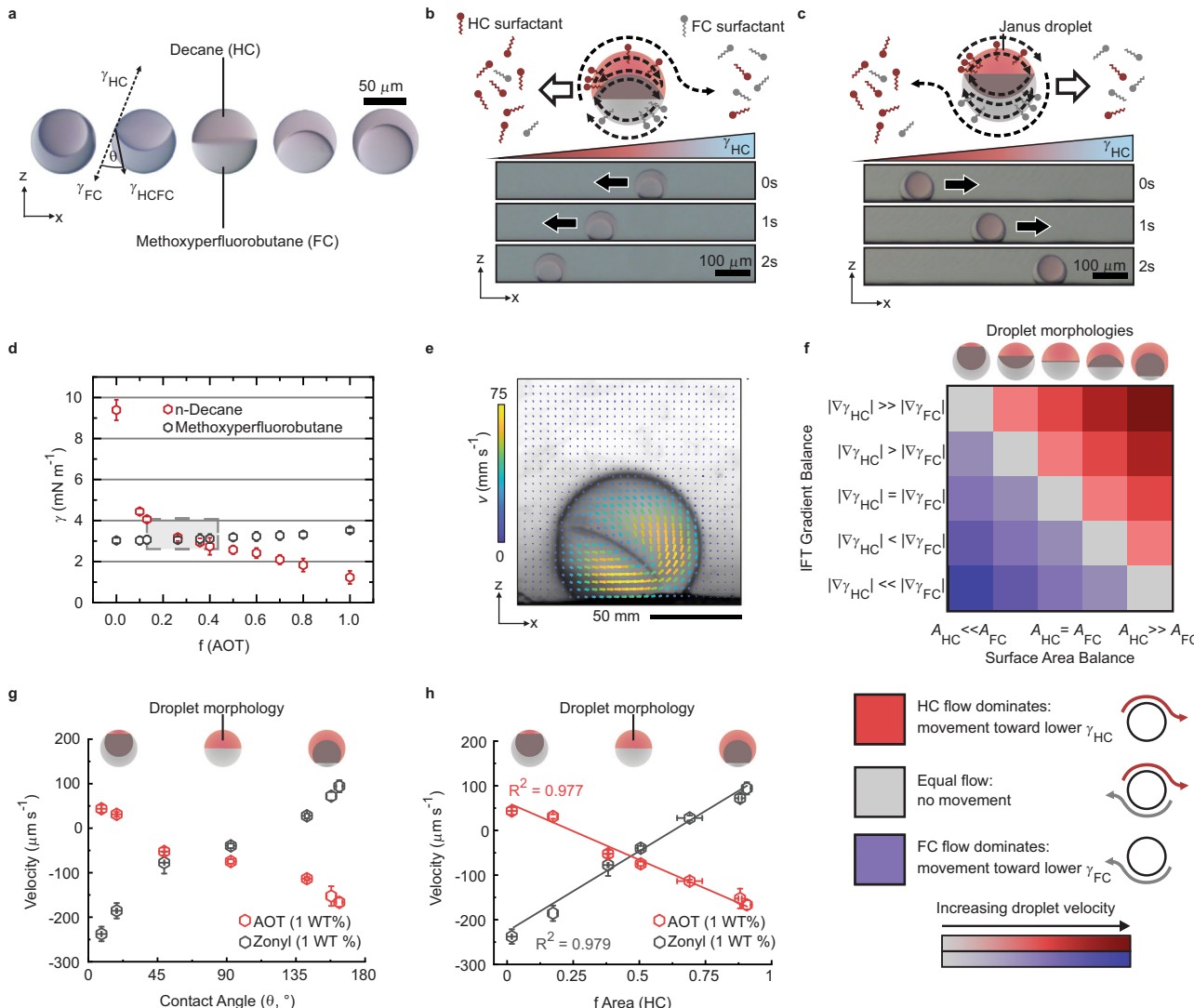

**Fig. 2 Reversible morphology-resolved chemotaxis of Janus emulsion droplets inside a laminar surfactant gradient. a** Side-view micrographs of Janus emulsion droplets in three different morphologies; the contact angle at the triple phase contact line is used to quantitatively describe the various Janus emulsion morphologies; **b** schematic illustration of the anisotropic Marangoni-type fluid flows surrounding a Janus droplet ($\theta = 155°$) inside a laminar surfactant gradient; side-view time-sequence micrographs of the Janus droplet moving toward the region where 1 wt.% of a pure AOT solution was added; **c** corresponding sketch and side-view time-sequence micrographs of the same Janus droplets in a different morphology ($\theta = 25°$) displaying movement away from the region where 1 wt.% AOT was added; **d** pendant drop tensiometry measurements of the individual droplet phases inside a 1 wt.% aqueous surfactant solution comprised of AOT and Zonyl in varying fractions, with the inset box reflection the region of dynamic droplet morphology reconfigurability; **e** particle image velocimetry (PIV) of flow fields surrounding Janus droplets in a hydrocarbon-dominant morphology responding to the addition of 1:9 AOT:Zonyl (Supplementary Movie 3); **f** conceptual sketch of the expected droplet chemotactic behavior based on a combination of the two independent factors, the respective interfacial tension (IFT) differentials across the two droplet interfaces and the droplet morphology; **g** morphology-dependent droplet velocities of Janus droplets moving in response to the addition of a 1 wt.% AOT and 1 wt.% Zonyl solution, respectively. A positive velocity notes movement away from the added surfactant, negative velocity notes movement toward the added surfactant; error bars reflect $n \geq 5$ measurements; **h** measured chemotactic droplet velocities in response to the addition of Zonyl and AOT plotted as a function of the HC surface area fraction.

$F_\mathrm{D}$ is the fluid–fluid Stokes drag force, $r$ the droplet radius, $\eta$ the viscosity of the continuous aqueous phase, and $\delta\gamma$ is the interfacial tension differential generated over the channel length $x$, which can thus be either positive or negative depending on the added probe (Eq. 1)[46].

Thus, a perfect Janus droplet ($\theta = 90°$) with equal surface areas ($A_\mathrm{HC} = A_\mathrm{FC}$) moves with respect to the dominant interfacial tension gradient across the HC interface ($F_\mathrm{HC} > F_\mathrm{FC} + F_\mathrm{D}$). With an increasing fraction of the HC surface area of the overall droplet surface area $f_\mathrm{HC} = A_\mathrm{HC}/(A_\mathrm{FC} + A_\mathrm{HC})$, the driving force for droplet displacement from fluid flows adjacent to the HC interface is more

dominant, increasing droplet velocities. In turn, when the area of the fluorocarbon interface is dominant ($A_\mathrm{HC} \ll A_\mathrm{FC}$), the overall force originating from the Marangoni flows adjacent to the FC interface can overcome the competitive force generated by the flows along the hydrocarbon hemisphere, despite the increased interfacial tension gradient across the hydrocarbon interface ($(|\nabla\gamma_\mathrm{HC}| > |\nabla\gamma_\mathrm{FC}|)$). Consequently, experimental observations reveal a decrease in droplet speed upon gradual decrease of the exposed HC surface area until at a contact angle of $\theta = 30°$, where droplets start to move in the opposite direction (Fig. 2g). Thus, both the respective interfacial tension gradients as well as the

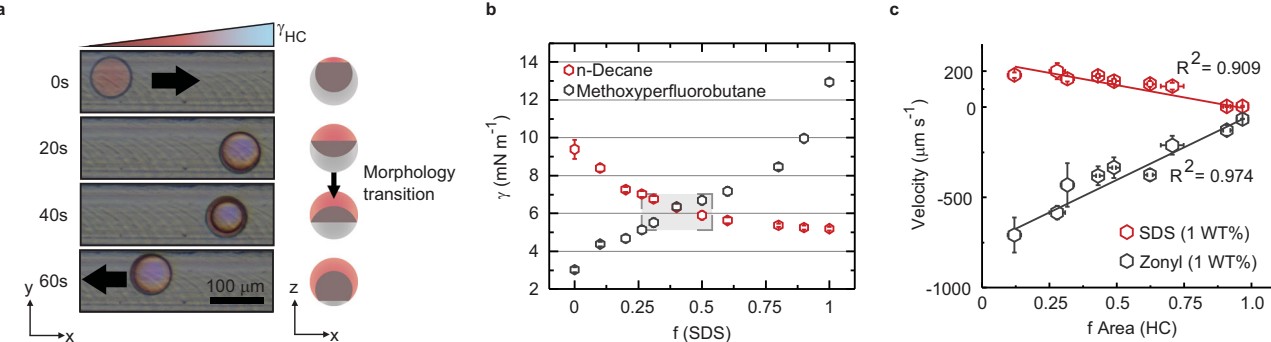

**Fig. 3 Adaptive chemotactic motile behavior of reconfigurable Janus droplets. a** Top-view time sequence micrographs of a Janus droplet with a fluorocarbon dominant morphology moving away from an added pure AOT surfactant solution nearing the point where the competitive Marangoni-type fluid flows on the upper and lower hemisphere of the droplets become equal, and the droplet becomes stationary. A surfactant-induced reconfiguration of the droplet morphology then leads to reversal of the direction of chemotactic motion as a result of the fluid flows adjacent to the hydrocarbon phase becoming dominant.; **b** pendant drop tensiometry measurements of the interfacial tensions of n-decane and methoxyperfluorobutane in overall 1 wt.% solutions of the surfactants SDS and Zonyl in varying ratios; **c** velocities of Janus droplets in different morphologies moving in response to the addition of a pure SDS ($R^2 = 0.91$) or Zonyl ($R^2 = 0.98$) solution plotted versus the ratio of exposed surface areas; error bars denote > 5 measurements.

droplet morphology mutually dictate the directionality and speed of Janus droplet motion, and Fig. 2f displays a conceptual overview of expected droplet motion resulting from a combination of the two independent factors. Physical considerations also reveal that in theory, for the case of a freely floating droplet, already very small differences in interfacial tension gradients across the two droplet interfaces ($|\nabla\gamma_{HC}| - |\nabla\gamma_{FC}|$) on the order of 25 mN m$^{-2}$, or synonymously, very small interfacial tension differentials on the order of $10^{-3}$ mN m$^{-1}$ across a single droplet suffice to induce a significant net fluid flow that is required to achieve droplet speeds of $v_{drop} > 100\ \mu m\ s^{-1}$. To further illustrate the morphology-dependent directionality and speed of chemotactic motion, we plotted the respective droplet velocities as a function of the fraction of HC surface area with respect to the droplet surface area (Fig. 2h). The resulting plot displays a linear correlation (Fig. 2h), illustrating the direct proportionality between the droplet velocity and the ratio of exposed surface areas.

The linear correlation between droplet velocity and surface area ratio is confirmed when the droplet velocity is tracked as a function of the exposed surface area ratio in an experiment, where we add the FC surfactant Zonyl as opposed to the HC surfactant AOT. Again, the majority of droplets in the various starting morphologies move towards the additional surfactant to minimize their overall surface energy. However, in this setup, droplets with a larger fluorocarbon surface area fraction move rapidly towards the added surfactant, whereas droplets with a larger hydrocarbon surface area fraction move away from the additional FC surfactant.

**Reversible and adaptive chemotaxis of Janus droplets.** In all experiments, the droplets respond to changes in surfactant composition with morphological reconfigurations in addition to chemotactic motion. Morphological reconfigurability is an inherent feature of Janus emulsions, which readily adapt their internal geometry to the force balance of interfacial tensions. However, these morphological changes occur at a much slower time rate when compared to the instantaneous onset of motion (>20 s; Supplementary Fig. 13) that is driven by the compensation of imbalances in interfacial tension across the droplet interface. In turn, the morphological change is a diffusion-controlled process, governed by the exchange of surfactant molecules at the droplet surface. To illustrate the time dependency of both and to exemplify the role of shape and interfacial tension difference in the chemotactic motion of Janus emulsions, we employ Janus

droplets in a morphology ($\theta = 25°$) close to the tipping point of reversed directionality of motion. In this system, when we introduce higher concentrated AOT solution to the microchannel, initial chemotactic motion was very slow due to the competitive fluid flows being almost in balance. Initially, Janus droplets move away from the oncoming surfactant solution, however, upon onset of the morphological reconfiguration, the droplet speed decreases until droplets reached a stationary state at which both flows on the upper and lower hemisphere are balanced. Further diffusion of the added surfactant solution then induces a continued change in droplet morphology resulting in an expansion of the hydrocarbon surface area. Consequently, droplets reverse their motion and start to move backwards in the direction of the original location (Fig. 3a and Supplementary Movie 4). The experiment reveals the adaptive nature of such Janus emulsion microswimmers, where multiple independent parameters, including the droplet composition, the surfactant type and ratio, as well as the internal droplet geometry mutually determine the directionality of the chemotactic motion.

To further elucidate the dependency of the droplet motion towards or away an oncoming surfactant concentration gradient we investigate a second surfactant combination, where we replace the hydrocarbon surfactant AOT with sodium dodecylsulfate (SDS, Supplementary Fig. 8). Contrary to the previous case, interfacial tension measurements reveal a significant dependency of both droplet phases on the surfactant ratio within the continuous aqueous phase (Fig. 3b). As a result of the increasing differences in net interfacial tension between Janus droplets and droplets in an encapsulated double emulsion morphology that were stabilized by a pure SDS or Zonyl solution, we anticipated pronounced Marangoni fluid flows induced by changes in surfactant composition. In fact, morphology-dependent measurements inside the microchannel reveal an increased droplet velocity towards the region of minimized surface energy. With the lowest interfacial tension of the emulsion droplets determined inside a pure Zonyl solution ($\gamma = 3.03$ mN m$^{-1}$) the induced chemotactic motion is dominated by the flows adjacent to the bottom fluorocarbon phase and consequently fastest droplet speeds are recorded for droplets in a fluorocarbon dominant morphology (Fig. 3c). A linear decrease of droplet velocities for droplets with an increased hydrocarbon surface area confirmed that the direction and speed of Janus droplet chemotactic locomotion is determined by the competitive anisotropic fluid flows surrounding the two opposite phases of the droplets.

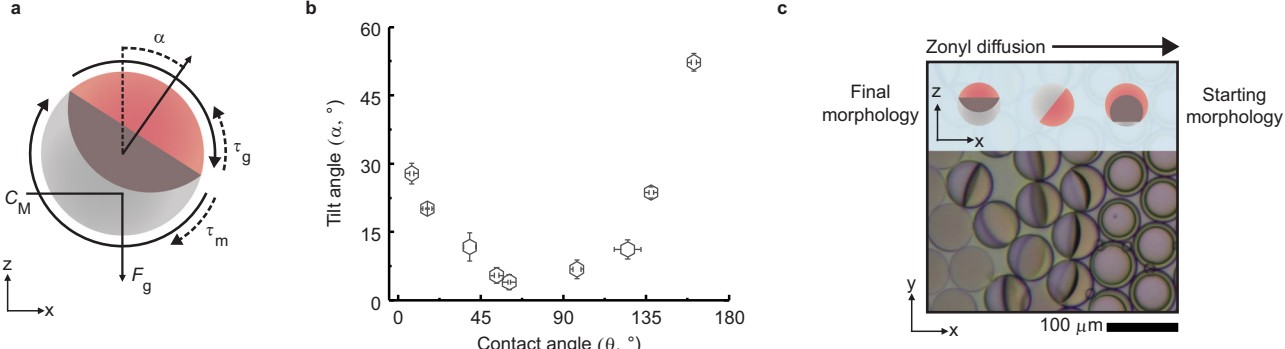

**Fig. 4 Janus droplet tilting caused by anisotropic fluid convections. a** Schematic diagram of Janus droplets tilting out of gravitational alignment in response to interfacial tension gradients across their interfaces that result in anisotropic Marangoni-type fluid convections. The Marangoni flow-induced torque $\tau_m$ is balanced by a gravitational torque $\tau_g$ due to the force of gravity ($F_g$) acting on the center of mass of the droplet ($C_M$), generating a tilt off the gravitational axis $\alpha$; **b** experimentally recorded Janus droplet (methoxyperfluorobutane:decane in AOT:Zonyl) tilting angles as a function of droplet morphology for Janus droplets inside a laminar surfactant gradient evoked by the addition of a 1 wt. % AOT solution to the microchannel, error bars denote > 5 measurements; **c** Inverted optical micrograph of a monolayer of monodisperse Janus emulsion droplets, originally stabilized by a 1 wt. % SDS:Zonyl (2.5:7.5) solution as a pure 1 wt. % Zonyl solution diffuses through the sample.

**Marangoni flow induced tilting of Janus droplets**. Notably, the Marangoni flows surrounding the Janus droplets also cause a rotation of the droplets out of their natural gravity aligned position, in addition to driving and directing their motion. This re-orientation of the droplets inside the laminar surfactant gradient can be explained with a surfactant gradient-induced torque $\tau_M$, balanced by a gravitational torque $\tau_g$ (Fig. 4a)[54]. To understand how the flow velocities on the two hemispheres affect droplet tilting, we measure the steady-state rotation angle α for Janus droplets in different morphologies after creating a surfactant gradient inside the microchannel (addition of 1 wt.% AOT). A pronounced torque is observed particularly when one of the two anisotropic fluid flows surrounding the opposing phases of the droplets is dominant and when the symmetry axis of the droplet was not aligned with gravity (Fig. 4b). Both the strength of the gravitational torque and the torque induced by the Marangoni flows depend on the droplet morphology. Specifically, due to the strong density contrast between the two droplet phases, the gravitational torque varies with the location of the center of mass $C_M$ of the droplets, which changes with droplet morphology. On the other hand, the power of the flow-induced torque depends on the respective strength of the fluid flows adjacent to the upper and lower hemisphere of the Janus emulsions and the net torque is determined by the difference in the respective flow velocities.

The fluid convection-induced droplet torque is also observed when reconfiguring droplet morphologies within a confined environment, such as in densely packed monolayers of droplets. In this instance, upon diffusion of a surfactant through a sample, droplets reconfigure in response to variation in the surfactant ratio. However, prior to this morphological reconfiguration, we observe a spontaneous actuation of droplet tilting. This effect can be explained by the surfactant-induced anisotropic Marangoni-type fluid convections surrounding the droplets caused by the oncoming surfactant gradient (Fig. 4c). Upon further diffusion of the surfactant and equilibration of the interfacial tension difference droplets reconfigure their morphology and relax back into their gravitationally aligned state. The latter experiment further showcases the temporal correlation between the immediate onset of Marangoni-type fluid convections in response to the interfacial tension differences across the droplet interfaces, and the surfactant diffusion controlled morphological reconfiguration of the droplets.

**Programmable chemotactic response of Janus droplets**. After validating the reversible and morphology-resolved chemotactic

actuation and motion of Janus emulsion droplets to externally triggered static surfactant gradients, we next explored the implications of the associated chemical equilibrium-driven and adaptive behavior of the emulsion droplet chemotaxis in response to stimuli-triggered changes in surfactant effectiveness. We were particularly interested in a stimuli-triggered evocation of surfactant gradients induced by physical or (bio-)chemical stimuli. To this end, we began by extending our investigations towards a switchable light-responsive azobenzene-based surfactant system (AzoTAB; Fig. 5a, Supplementary Fig. 9) that can undergo a reversible photo-induced *cis–trans* isomerization depending on the incident wavelength[55]. Employing an external focused light beam allowed to photo-switch the thermodynamically more stable *trans*-isomers locally into the bent *cis*-form via UV-light irradiation ($\lambda = 365$ nm), inducing a local change in the hydrophobicity of the surfactant and thus a decrease in its surfactant effectiveness[33]. This switch is reversible with the application of blue light ($\lambda = 470$ nm). Non-uniform delivery of UV light results in a light-induced gradient in hydrocarbon surfactant effectiveness across a sample (Supplementary Fig. 10). Janus droplets with an hydrocarbon-dominant morphology responded by moving away from the UV light spot towards regions of increased concentrations of the AzoTAB surfactant in its more effective *trans* configuration. In these experiments, the motion was sustained until the surfactant concentration across the sample equilibrated, together with a morphological reconfiguration of the droplets across the sample (Fig. 5b; Supplementary Movie 5).

To generate a more persistent surfactant gradient across a droplet sample, we irradiated the whole setup permanently with blue light, while applying the UV-light only to a focused spot, thereby triggering an immediate re-isomerization of the AzoTAB surfactants upon diffusion out of the UV irradiated area (Supplementary Movie 6). Here, localization of the UV-light beam in the vicinity of the Janus droplet in a fluorocarbon dominant morphology led to a continued chemotactic motion towards the light beam (Supplementary Fig. 11). In this setup, the motion in response to the photochemically triggered gradients in surfactant composition was dominated by the Marangoni-type fluid flows at the fluorocarbon interface that were directed away from the light source, whereas the flows adjacent to the hydrocarbon phase pointed into the opposite direction. By dispersing particles into the aqueous continuous phase, we were able to track and confirm these anisotropic fluid flows (Supplementary Movie 7)[56]. Substantial anisotropic fluid movement was perceived (Supplementary Fig. 10)

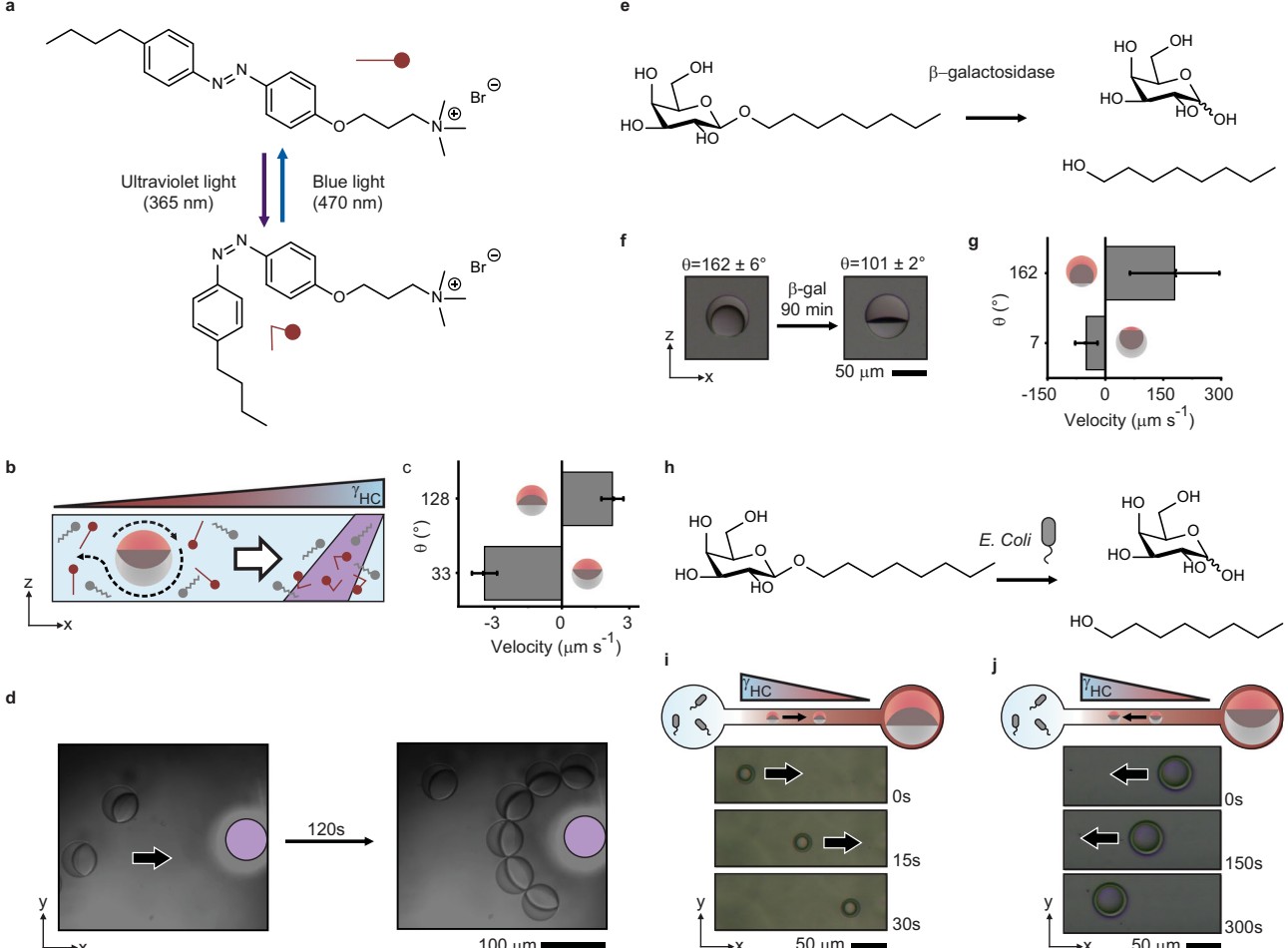

**Fig. 5 Chemotaxis of Janus droplets in response to photo- and bio-chemically induced surfactant gradients. a** Representation of the photo-triggered reconfiguration of an AzoTAB surfactant in response to the application of UV (365 nm) and blue (470 nm) light; **b** schematic sketch displaying the effect of local UV light application in the vicinity of AzoTAB stabilized Janus emulsions; **c** velocity of decane:methoxyperfluorobutane droplets in AzoTAB:Zonyl versus the contact angle of the starting droplet, which respond to the gradient generated by AzoTAB and move away from the UV light spot dependent on starting morphology under blue irradiation, error bars denote > 5 measurements; **d** optical micrographs displaying a self-assembly of Janus droplets around a localized UV light spot (indicated) under bright field blue light irradiation (Supplementary Movie 8); **e** chemical scheme for enzymatic ($\beta$-galactosidase) cleavage of a galactose-based hydrocarbon surfactant; **f** decane:methoxyperfluoroubutane droplets stabilized in 0.5 WT % $\beta$-n-octyl-galactopyranoside and 0.05 WT% Zonyl retain a hydrocarbon-dominant morphology ($\theta = 162°$). After the addition of 3U mL$^{-1}$, the droplet morphology reconfigures to $\theta = 101°$ as the hydrocarbon surfactant effectiveness is weakened by enzymatic cleavage of the hydrocarbon surfactant; **g** velocity of complex droplets composed of decane and methoxyperfluorobutane in fluorocarbon-dominant (0.5 WT% $\beta$-n-octyl-galactopyranoside, 0.1 WT% Zonyl, $\theta = 7°$) and hydrocarbon-dominant (0.5 WT% $\beta$-n-octyl-galactopyranoside, 0.05 WT% Zonyl, $\theta = 162°$) droplets to the addition of 10 µL of their stabilizing mixture containing 3 U mL$^{-1}$ $\beta$-galactosidase, error bars denote > 5 measurements; **h** chemical scheme for an extracellular enzyme-mediated cleavage of the same surfactant upon addition of E.coli bacteria; **i** Janus droplets in a hydrocarbon-dominant morphology display chemotactic movement selectively away from the E. Coli colonies. **j** Due to the enzyme-mediated surfactant cleavage, Janus droplets in a fluorocarbon-dominant morphology display chemotactic movement selectively towards the added E. Coli bacteria.

and served to explain the flow-induced tilting of the droplet out of its gravitational alignment. The dominance of the flow adjacent to the hydrocarbon phase of the droplet upon triggering a reconfiguration of the respective hydrocarbon surfactant was confirmed, explaining the motion of the droplets towards the UV-light source only for Janus droplets in a fluorocarbon dominant morphology. These experimentally observed flow patterns verified the underlying significant role of the fluid flow anisotropy as the main reason for the directional chemotactic actuation and motion of the Janus droplets in different starting morphologies. With the directionality of the Janus droplet chemotactic motion being controllable with their starting morphologies, such a setup was employed to initiate a light-directed patterning of droplet networks (Fig. 5d). Illumination of the entire

setup with blue light, resulted in droplets experiencing a permanent surfactant gradient surrounding the UV light spot, which serves as an optical trap for a targeted placement, actuation of tilting, and assembly of the Janus droplets.

Beyond a light-triggered evocation of a surfactant gradient, we next investigated a chemically induced local variation of the surfactant effectiveness. A cleavable galactose-based hydrocarbon surfactant ($\beta$-n-octyl galactopyranoside; Fig. 5e) serves as an example for a biochemically triggerable surfactant system, mediated by the enzyme $\beta$-galactosidase. Droplets in a hydrocarbon dominant morphology stabilized by a mixture of this surfactant and Zonyl respond to an addition of $\beta$-galactosidase with morphological transitions towards a more 'opened-up' Janus morphology (Fig. 5f). This behavior is explained by the

dramatically altered surfactant effectiveness as a result of the enzyme-mediated cleavage of the galactose amphiphiles stabilizing the hydrocarbon-water interface into the respective hydrophilic and hydrophobic parts, as also confirmed by NMR experiments (Supplementary Fig. 12). The addition of the enzyme $\beta$-galactosidase-containing surfactant phase (3 U mL$^{-1}$) to the open end of the droplet containing microchannel caused a temporary surfactant gradient within the droplet surrounding continuous phase. Depending on their starting morphology, droplets respond to the interfacial tension differential with a chemotactic motion directed selectively towards or away from the region of $\beta$-galactosidase addition. Enzyme action decreases hydrocarbon surfactant effectiveness such that droplets with a hydrocarbon dominant morphology moved away from the area of decreased hydrocarbon effectiveness, whereas droplets with a fluorocarbon dominant morphology readily approach the region where the enzyme solution was added (Fig. 5g). The same droplet response is then observed also upon the addition of Escherichia coli (E. coli) bacteria to the microchannel. E. coli dispersions, as many other bacteria, contain strain-specific mixtures of extracellular enzymes and characteristically produce the extracellular enzyme $\beta$-galactosidase, which is commonly targeted in biorecognition assays[57]. Side-selective addition of the bacteria (10$^8$ CFU mL$^{-1}$) to the microchannel resulted in directional chemotactic interaction of the Janus droplets with E.coli bacteria (Fig. 5i, j and Supplementary Movie 9). Local changes in surfactant composition via an enzyme mediated decrease of the hydrocarbon effectiveness surrounding the bacteria colonies resulted in selective chemotactic motion of droplets with a hydrocarbon dominant morphology away from the bacteria. In contrast, droplets with a fluorocarbon dominant morphology approached the bacteria. The motion lasted for several minutes until a diffusion of the bacteria through the microchannel and consequently an equilibration of the hydrocarbon surfactant concentration across the sample resulted in deceleration of the droplet motion along with morphological adaptations.

## Discussion

Janus emulsions comprised of two hemispherically aligned phase-separated fluids show a controllable and reversible chemotactic movement up or down an oncoming surfactant gradient. The directionality of the chemotactic response depends on the internal morphology of the droplets and can be dynamically altered. The basis of the phenomenon is that the two opposite interfaces of Janus droplets experience different interfacial tension differentials inside a laminar surfactant concentration gradient within the continuous phase. As a result, two different competitive Marangoni-type fluid flows develop along the upper and lower interface of the gravitationally aligned fluid phases. The overall droplet motion with respect to the surrounding liquid, i.e. both speed and directionality, is determined by the respective velocities of the two competitive flows that are directly proportional to the differences in the respective interfacial tension gradients as well as the surface area ratio of the exposed interfaces as defined by the Janus droplet morphology.

While isotropic single-phase emulsion droplets display an exclusively unidirectional motion towards lowering their interfacial tension when placed inside a laminar surfactant gradient, the introduction of permanent asymmetry to the internal droplet geometry and composition, combined with the ability to reversibly adjust the internal droplet morphology, opens a path towards a continuous and reversible modulation of the directional chemotactic response of moving droplet systems. Utilizing an external evocation of a laminar surfactant gradient inside a

microchannel, we have shown that the direction of a Janus droplet chemotactic motion, as well as their velocity, can be predictively fine-tuned. Combined with the dynamic reconfigurability of Janus droplets, we obtain a highly adaptive material system that autonomously responds to marginal changes in the chemical environment. For example, induced by small interfacial tension gradients, droplets can initially move into one direction, and subsequently, due to a surfactant-induced change in their internal configuration, move into the opposite direction. The highly dynamic and reversible programmability of the Janus droplet localization was further illustrated in experiments where a chemical gradient was evoked by light or biochemical triggered local changes in surfactant effectiveness. Thereby, a droplet system was realized that displayed programmable motion towards or away a certain chemoattractant or—repellent, such as bacteria cells, yielding a novel type of adaptive, entirely fluid-based microswimmer with directional propulsion profiles. We have also shown that in addition to chemotactic movement, anisotropic fluid currents surrounding the droplets also actuate a temporally evoked tilting of the Janus emulsions inside a surfactant gradient.

Complex droplets are widely employed in encapsulation and release, dynamic imaging, medical, pharmaceutical, and sensing platforms and we expect that a combination of the chemotactic actuation and motile characteristics with the unique chemical, morphological, and optical properties of this material class will pave the way towards the generation of a new class of adaptive, autonomously operating, chemo-intelligent soft matter colloids with strong implications for a series of applications, including microrobotics, dynamic optical components, advanced biosensors, and for the study of emergent properties arising from interacting out-of-equilibrium systems.

## Methods

**Chemicals and biologicals**. The following materials were used as received without further purification: n-Decane (Sigma Aldrich, 99%), methoxyperfluorobutane (ABCR, 97%), Sudan Red 7B (Sigma Aldrich), di-n-octyl sulfosuccinate sodium salt (Sigma Aldrich, 98%), Zonyl FS-300 (ABCR), sodium dodecyl sulfate (Sigma Aldrich, 98%), $\beta$-n-octyl-galactopyranoside (Combi-Blocks), Escherichia coli (E.coli DH5 $\alpha$- New England Biolabs), $\beta$-Galactosidase from Aspergillus oryzae (Sigma-Aldrich). Bacteria experiments were performed in PBS buffer solution (pH: 7.2). DI water was used for the emulsion continuous phase preparation.

**Droplet generation**. Monodisperse biphasic droplets were generated with a thermal phase separation approach in microfluidics[49]. A hydrophilic "X" junction Dolomite glass chip was employed with different continuous phase/dispersed phase pressure ratios to produce droplets with a diameter of ~50–100 μm. Fluid flow rates were controlled with a Fluigent Flow-EZ pressure control platform. Janus droplets were generated above the UCST of the respective oil mixture in a custom-made thermal chamber. Specifically, decane and methoxyperfluorobutane, were heated above their UCST (28 °C) prior to emulsification. Droplets were then generated in a surfactant mixture for the required experimental probe, and cooled to induce phase-separation, generating Janus droplets in uniform internal droplet morphologies. For SDS:Zonyl experiments, the droplets were emulsified in a mixture of 4:6 1 WT % SDS:Zonyl. For AOT:Zonyl experiments, the droplets were emulsified in a mixture of 0.358:0.642 1 WT % AOT:Zonyl. For AzoTAB:Zonyl experiments, droplets were first generated in a 4:6 1 WT % SDS:Zonyl solution, and then phase transferred twice into the respective AzoTAB:Zonyl solutions in the desired ratio. As-formed droplets were redispersed twice into solutions of different surfactant compositions to yield Janus emulsion in the desired internal droplet morphologies. Droplets morphologies were generated based on calibration curves, generated from sideview micrograph contact angles of generated droplets, for a given surfactant combination (AOT:Zonyl in Supplementary Fig. 7, SDS:Zonyl in Supplementary Fig. 8, or AzoTAB:Zonyl in Supplementary Fig. 9). Generated microfluidic droplets were stored in sealed, cooled containers, and used within 3 days. Droplets generated were verified for volume ratio and or contact angle variation before any experiment.

**Optical microscopy**. Side-view time-sequence micrographs and videos of complex droplets were collected using a custom-built microscopy setup, composed of a 200 mm tube lens (Thorlabs), planar optical microscopy lens (4×, 10×, or 20×, Olympus), and area scan CCD camera (HIKvision). For anisotropic light delivery,

530 nm light (LED, Thorlabs) was collimated and delivered to the sample. 365 nm light (LED, Thorlabs) was collimated, focused with an objective to a point, and delivered to the sample on an XYZ stage (Thorlabs) for translation and focus of the light. 470 nm light (LED, Thorlabs) was collimated and delivered to the whole sample (also mounted on a Thorlabs XYZ stage). This custom setup was mounted vertically to study movement effects due to this light (Supplementary Fig. 10). Inverted microscopy was performed on a Bresser IVM 401 microscope equipped with an area scan CCD (HIKVision).

**Contact angle determination**. Contact angles of gravity-aligned complex droplets at their three-phase contact line were used to measure and quantitatively describe their morphology. The radius of the droplet and the radius of the internal curvature, and the distance between these two points were measured (Supplementary Fig. 4). With this, the contact angle, as well as the balance of droplet surface areas could be determined based on the Neumann construction (Supplementary Fig. 4) and the law of cosines. Due to optical effects between the droplet interfaces, a correction factor was applied based on the refractive indices of the droplet phases to the radius of the internal curvature and the distance between the droplet radius and the curvature centerpoint where $R_{real} = n_{medium}/n_{outer}R_{image}$. All droplet contact angles, surface areas, and volume ratios were calculated from the diameter of the droplet, the diameter of the internal curvature, and the distance between the centerpoint of the droplet and the internal curvature as determined from side-view optical micrographs using imaging processing software (Fiji).

**Pendant drop tensiometry**. Pendant drop tensiometry was performed on a Krüss DSA10-MK2 drop shape analyzer. Needle-attached droplets were imaged, fit, and analyzed by Krüss Advance software via Laplace equation. Measurements were taken after exponential decrease post droplet-formation, due to surfactant adsorption, or after interfacial tension values lowered at a rate of <0.1 mN m⁻¹ min⁻¹, and the curve fit to find the equilibrium interfacial tension[49]. Surfactant solutions were prepared as fractions, and in particular for surfactant combinations to produce particular droplet geometries. Needles and cuvettes were cleaned with water and acetone and dried between measurements.

**Channel-directed droplet movement**. To realize a concentration-gradient-based droplet movement, induced by added concentrated surfactant solution, a milled polycarbonate (for alkane resistance) channel was used (Supplementary Fig. 1). The cell had a channel width of 300 μm and height of 300 μm, and a cylindrical droplet well on either side of the channel which was 5 mm in diameter and 1 mm deep. Droplets were placed in a 1 mm deep well of the channel, and the channel filled with the starting surfactant, before the chip, including wells and channels were closed with a standard glass slide and inverted on a microscope, enabling the droplets to sit on the glass slide. The droplets were allowed to settle, and once still, the desired experimental condition was applied via liquid addition to an open end of the channel. Various surfactant solutions were added to the open end of the channel to study the movement as a result of the interfacial tension gradient. For this, 10 μL of surfactant solution was pipetted onto the access port of the channel, and the resulting movement recorded with video. Velocities of singular, similarly-sized, and freely moving droplets were recorded with droplet tracking, or if droplet movement approached or exceeded 1 droplet diameter per frame, velocity was measured manually. Droplet velocity sequences were undertaken with microfluidic droplets prepared as-per specification, and experiments undertaken with the same droplets, of same radius, on the same day to ensure experimental comparability e.g. due to uneven evaporation of one droplet phase altering the volume ratio and therefore the droplet surface areas. Between experiments, the chip was cleaned 3× each with water and ethanol, and dried.

**Droplet tracking methods**. Droplet videos were collected at various frame rates and analyzed with a modified MATLAB script utilizing the particle-tracking code of Crocker and Grier to droplets with the use of in-built circle detection in MATLAB[48]. Videos were analyzed per frame, and tracked droplets were then analyzed for distance moved per frame or unit of time. Due to the linearity of movement in the channel, droplet movement in the x–y plane was stable. Unimpeded, freely moving, independent droplets were analyzed for their velocity. Due to the ultimate reconfiguration of droplets as diffusion of the surfactant-probe completed throughout the channel, droplet speeds in response to the interfacial tension gradient were recorded toward the beginning of the droplet movement after initial acceleration. For droplets where velocity was larger than half the radius per frame, droplet velocity was measured manually.

**Synthesis of the light-responsive surfactant AzoTAB**. The light-responsive surfactant 4-butyl-4′-(3-trimethylammoniumpropoxy)phenylazobenzene (AzoTAB) was synthesized according to a previously reported procedure[51]. In brief, at first, 4-butyl-4′-hydroxyazobenzene was synthesized via diazocoupling between 1 eq. 4-butylbenzenediazonium chloride and 1 eq. phenol. The 4-butylbenzenediazonium chloride was generated in situ via reaction of 1 eq. sodium nitrite with 1 eq. 4-butylanilinin. Next, the propyloxy spacer was introduced via substitution of 1,3-dibromopropane (1.1 eq) by the phenate (1.eq). Finally, the quarternized ammonium head group was introduced via dropwise addition of trimethylamine (4 eq.) to

1 eq. of 4-butyl-4′-(3-bromopropxy)azobenzene under reflux conditions to yield 4-butyl-4′-(3-trimethylammoniumpropoxy)phenylazobenzene in 19% yield.

**Light-triggered droplet movement**. Movement of droplets as a result of induced-interfacial tension gradients generated by the photo-switchable surfactant AzoTAB were performed in the experimental setup described in Supplementary Fig. 10. Decane:methoxyperfluorobutane droplets stabilized by AzoTAB:Zonyl (0.23 WT% AzoTAB 0.1 WT% Zonyl, $\theta = 127°$) in a hydrocarbon-dominant morphology were placed in a level sample holder, to which the entire sample had delivery of blue (460 nm light). On addition of UV light (365 nm), an interfacial tension gradient was generated, generating Marangoni flows across the surface of the droplet and thus movement toward the area of higher interfacial tension (away from UV light). Droplets were also prepared in AzoTAB:Zonyl (0.1 WT% AzoTAB 0.1 WT% Zonyl, $\theta = 32°$) with fluorocarbon-dominant morphologies before the application of UV and Blue light to the sample.

**Particle image velocimetry**. For particle tracking methods, Particle Image Velocimetry was calculated using the tool from Thielicke and Stamhuis, utilizing videos with added tracer particles[33]. Videos were recorded at 20–100 frames per second. Immobilized droplets in AzoTAB:Zonyl were placed in a sample holder and had anisotropic light applied to induce an interfacial tension gradient, and monitor the resulting flow fields in side-view microscopy (Video S6). Droplet bodies were separated from the solution in-software to analyze only movement of freely-moving tracer particles. Velocity flows were calculated in-software with the FFT multi-pass with varying integration areas based on the frame resolution (256, 128, 64). This information was corrected for scale, and averaged over the timestep and velocity analyzed.

**Biochemically triggered droplet chemotaxis**. Monodisperse droplets generated in 4: 6 1 WT% SDS: Zonyl were twice-phase transferred into 0.5 WT % β-n-octyl-galactopyranoside with varying Zonyl concentrations (0.05 WT% for droplets with ($\theta = 162°$), and 0.1 WT% Zonyl for droplets with ($\theta = 7°$). An identical 'probe' solution to the droplets was prepared with the addition of β-galactosidase (3 U mL⁻¹), and gently stirred at room temperature for 90 min. At this time, observation of the decreased hydrocarbon-surfactant effectiveness was observed with sideview microscopy, lowering the contact angle of droplets from $\theta = 162°$ to $\theta = 101°$. Furthermore, to observe the induced droplet velocity as a result of this surfactant gradient, the addition of either droplet phase's incubated, enzyme-containing surfactant mixture (10 μL) to droplets within the microfluidic chip was undertaken and the droplet response recorded.

**Droplet movement in response to *E. Coli***. Microfluidic droplets generated in 4:6 1 WT % SDS: Zonyl were twice phase-transferred into a solution containing 0.5 WT % β-n-octyl-galactopyranoside and 0.05 WT % Zonyl. All experiments were conducted at room temperature. E. coli (DH5 α- New England Biolabs) was incubated in lysogeny broth from frozen glycerol stock overnight at 37 °C until a late-logarithmic phase culture was obtained, diluted to 0.6 OD, and further grown to 1.0 OD which corresponded to $8 \times 10^8$ CFU mL⁻¹. The presented microfluidic chip (Supplementary Fig. 1) was prepared with droplets in one end of the channel, sealed, and allowed to rest on an inverted microscope. 1 μL of E. coli at $8 \times 10^8$ CFU mL⁻¹ was carefully inserted into the channel at one end of the channel (total volume ~159 μL) and droplet movement observed after rest. For the purposes of our experiment, we found that 0.5 WT % β-n-octyl-galactopyranoside: 0.05 WT % Zonyl with phase transferred decane:methoxyperfluorobutane droplets generated 'open' hydrocarbon-dominant complex droplets, and 0.5 WT % β-n-octyl-galacto-pyranoside: 0.1 WT % Zonyl generated decane:methoxyperfluorobutane droplets with a 'closed' fluorocarbon-dominant morphology.

## Data availability

Data which supports the findings of this study are available from the corresponding author upon request.

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

## Acknowledgements

The authors gratefully acknowledge financial support from the Max-Planck Society and from the Emmy-Noether program of the German Research Foundation under grant no. ZE 1121/3-1 (L.Z.). We are grateful to the MPIKG mechanical workshop for their help with chip manufacturing.

## Author contributions

B.D.F., P.G., and L.Z. conceived the project; P.H.S and L.Z. supervised and supported the project; B.D.F. designed the experimental setups and performed all droplet motion experiments, for which P.G. and L.Z. provided guidance; S.D. synthesized the light-sensitive surfactant; A.W.B. studied enzyme-sensitivity of Janus emulsions; A.W.B. and P.H.S. supported experiments monitoring chemotaxis of droplets in response to bacteria; B.D.F. and L.Z. analyzed the data; all authors discussed the results and contributed to the writing and editing of the manuscript.

## Funding

## Competing interests

The authors declare no competing interests.
