## [Peer review file · Nature Communications]

REVIEWER COMMENTS

Reviewer #1 (Remarks to the Author):

It is very hard for me to detect where there is any kind of novelty inside this manuscript. The authors presented some active Janus oil droplets in water. The physical discussion is missing, there is not even a qualitative description of the active motion, no flow field characterization via PIV, which is strange as the paper is centered on the motion of active Janus droplets.

I don't see the novelty concerning the usage of photoactivable surfactant, as there are many similar articles like: https://www.baigllab.com/docs/PDF/Diguet_Photomanipulation.pdf.

I don't see where using an active Janus is an advantage anywhere here.

I have the same comments concerning the design and control of the flow motion of active Janus droplet with similar perfluorated surfactant: as a much more detailed article was published online recently, including a much more detailed flow control description and chemical description (Meredith, C., et al. Chemical design of self-propelled Janus droplets. Preprint at <https://doi.org/10.33774/chemrxiv-2021-p7jqv-v2> (2021). REF37 in the manuscript).

Also, the chemical composition used here, prevents any kind of bio applications of these Janus active droplets, which is another strong limitation.

The article redaction is moreover confusing as many infos are put in suppmat but discussed in detail into the main text, which made this article not suitable for a broad audience (see the mention to figure S3 page 6.)

Also, I have a big doubt concerning the reality of the results concerning the measured active Janus droplet velocities present in suppmat video 1 and 2. These videos seem to show a laminar driven flow BEHIND and not a purely active flow. In particular, an extremely similar other system in chemical composition (the one in ref37) presents Janus droplets velocities that were 1 order slower (even nearly 2 orders), which looks a confirmation to me that the authors did not perform the experiments correctly and measured the motion of active droplets into a strong driven flow.

Behind these criticisms, I only detect two interesting points. One the droplet active motion is extremely fast, however the reason is not explained by the authors. I believe these results came from experiments

that were not performed correctly. Also, the part using the enzymatic reaction is interesting but also not really novel... for an unclear reason, it is even barely discussed by the authors.

Reviewer #2 (Remarks to the Author):

Authors report the chemotactic actuation and motion of Janus emulsion droplets that respond to surfactant gradients. The work is original, and the story is scientifically convincing. I am impressed by their strong efforts, e.g., their thorough datasets in SI and the experiments with light, chemicals, and bacteria, to demonstrate the implications of the chemotactic behavior of the Janus emulsion droplets. I believe this work would be of interest to the soft matter community. I want to recommend the publication of the manuscript in Nature Communication, only after improving some drawbacks.

1.

I have had difficulties understanding some details of the work. It is partly because of their mistakes in manuscript writing. To name a few, the caption does not match the plots in Figure 2 and Figure 5. Line 387 says the wavelength is 460 nm but 470 nm in Figure 5. I am also afraid that some choices of wordings are rather clumsy. For example, in line 395, I presume "continuously" is better than "permanently." I have spotted similar phrases/sentences/mistakes throughout the manuscript.

2.

The authors highlighted the "intrinsic out-of-eq" nature of emulsion droplets in the abstract. I agree, as written in line 60, the emulsion droplets are at a meta-stable state. I want to ask the authors what "out-of-eq" aspects of the emulsion droplets make them a suitable model system. For instance, the exchange of the surfactant molecules between the interface and continuous phase is just a dynamic 'equilibrium,' not "out-of-eq," isn't it? Please enlighten readers and me.

3.

Because the gravitational alignment is not mentioned in the abstract, the upper and lower hemisphere (line 24) sound awkward.

4.

I wonder what the authors mean by "permanently asymmetric" in line 88. Isn't Janus colloid (e.g., catalyst-coated) permanently asymmetric?

5.

I believe the authors should be cautious when they mention the interfacial tension gradient. They have characterized their system by interfacial tension 'differential' (difference in the interfacial tensions at two ends), not by the gradient, which is the difference divided by the channel length (20 mm according to the SI), i.e., the slope, if the tension changes linearly. What determines the speed is not the difference but the gradient. Specifying the gradient would be essential to compare the authors' data with other experiments. Furthermore, $f(\text{AOT})$ in plots is a handy parameter for the experiments but does not give direct information about the difference and gradient; I see Figure S3 gives some details.

5.5

After dropping the surfactant solution to the one end of the channel, how long does it for the channel to attain the linear γ gradient? How can we tell?

6.

The interfacial tension values from the tensiometry may need more info about its accuracy/precision. They are from the fitting of the interface, and the authors claim the roles of its subtle difference in line 186.

7.

I have trouble understanding how and why MPFB droplets move away from the region of high HC surfactant concentration. The authors give the interfacial tensions inside pure AOT and Zony and argue they have a small difference. Specifying the situation, i.e., the initial background type/concentration of surfactant and the tension gradient, may help.

8.

Showing the axes (x,y,z) or the direction of gravity will be helpful. For instance, in Figure 2 b&c. In the same vein, the side-view schematic seems quite different from the experimental side-view image, which

is almost in an engulfed configuration. Can the authors comment on this? Additionally, is the authors' Marangoni convection scheme still valid when they are in the engulfed configuration?

9.

I recommend the authors should think again about the meaning of Figure 2e. With the fixed volume ratio of the two phases, the surface area ratio is geometrically related to the contact angle; they can be calculated analytically with no help from experiments.

10.

To control the starting droplet morphologies, the authors dispersed the droplets into the continuous phases of different surfactant concentrations. These different initial concentrations should affect the gradient when the authors add the surfactant solution to the other end of the channel. Can the authors comment on this?

11.

The authors should be more quantitative about the time scale of the morphology change, instead of "much slower" in line 281. Does the droplet maintain the same morphology (and the resulting same speed) when traveling along the gradient? In a similar vein, f_{Area} in Fig. 3c is the area fraction in the starting morphology?

12.

Fig. 4c needs more explanation. What configuration is changing into what configuration? In the figure, what is what? I cannot discriminate the tilted one from the reconfigured one.

13.

What is the meaning of "a mechanically confined" in line 405?

Reviewer #3 (Remarks to the Author):

This paper describes the migration of biphasic Janus droplets in surfactant gradients. Homogeneous (non-Janus) drops move to regions of lower interfacial tension (high surfactant concentration) via Marangoni flows at the drop interface. The Authors demonstrate how drops of different liquids can move in different directions when positioned in a common gradient containing two surfactants. This observation provides a basis for creating spherical Janus droplets that can move up or down surfactant gradients depending on their internal morphology. The Authors show how the Janus morphology can change over time thereby reversing drop motion in the gradient. In addition to drop migration, the Authors quantify the gradient-induced rotation of Janus droplets from their preferred gravitational alignment. The Authors explore other routes for generating gradient-induced motions using reactive surfactants based on photoisomerization and enzymatic cleavage.

The use of Janus morphology to tune the gradient-induced migration of droplets within environmental gradients is an interesting new capability. The fact that this internal degree of freedom is also responsive to the environment has implications for designing autonomous behaviors based on internal feedback mechanisms (see, for example, Alvarez & Isa, *Nat Commun*, 2021, doi:10.1038/s41467-021-25108-2). The paper is very clearly written, and the conclusions are well supported by high quality experiments. In hindsight, the results are not particularly surprising; however, the experimental realization of this concept is non-trivial and merits publication in *Nature Communications*.

The Authors should cite related work on Janus droplets by the Zarzar group (Chemical Design of Self-Propelled Janus Droplets, doi: 10.26434/chemrxiv.14378780.v1).

Point-by-point response:

Reviewer #1: *It is very hard for me to detect where there is any kind of novelty inside this manuscript. The authors presented some active Janus oil droplets in water. The physical discussion is missing, there is not even a qualitative description of the active motion, no flow field characterization via piv, which is strange as the paper is centered on the motion of active Janus droplets.*

- We appreciate the reviewer's detailed and critical feedback, and are grateful for the time spent reviewing our manuscript. We are convinced that the concerns and issues raised by the reviewer helped to significantly improve the presentation of our findings in the revised version of our manuscript. With regard to the mentioned general concern about the novelty of our findings, we would like to begin our response by addressing an apparent central communication error present within our originally submitted version of our manuscript:

In our study, we reveal, for the first time, a directional, reversibly alterable, adaptive chemotactic response of emulsion droplets to an oncoming chemical concentration gradient. While chemically evoked interfacial tension differentials across emulsion droplet surfaces have been widely employed previously to induce and drive a unidirectional chemotactic motion of emulsion droplets towards regions of lowered interfacial energy, the chemotaxis behavior of multiphase, i.e. complex emulsion droplets has not been investigated in this context. Generally, for a chemotactic droplet movement that occurs due to Marangoni-type fluid flows generated at the interface between droplets and the continuous phase there are two separate main propulsion mechanisms reported in the literature:

- i) An interfacial tension differential across a droplet can be evoked through an anisotropic micellar solubilization of the dispersed phase. Here, gradients in the surfactant density at the droplet interface caused by dispersed phase solubilization into surfactant micelles creates interfacial tension gradients and propels the drops via the Marangoni effect, which is the mechanism underlying the motion of droplets in reference 37 (now 39) mentioned by the reviewer (Meredith et al, Matter, 2022). Higher concentrations of solute-“filled” micelles, associated with higher interfacial tensions, cause droplets to move towards regions with more “empty” micelles and thus towards regions of lowered interfacial energy. As a result, in this mechanism, droplet motion can be sustained and droplets become self-propelled due to repulsion from their own swollen micelle trail when hydrodynamic instabilities result in spontaneous symmetry breaking of the fluid flows surrounding the droplet. Droplet velocities up to 400 $\mu\text{m/s}$ have been reported (Hirono et al. *Langmuir*, 2018) for this propulsion mechanism.
- ii) Alternatively, passive surfactant-stabilized droplets can be propelled chemically, via local changes in the surfactant composition, strength, or effectiveness (e.g. via chemical reactions with responsive surfactants). Interfacial tension differentials across a droplet interface evoke convective Marangoni-type fluid flows from regions of low interfacial energy to regions of high interfacial energy that drive the droplet locomotion (a mechanism our work is built upon). Very small interfacial tension imbalances suffice to achieve remarkable droplet velocities and

experimental reports realized droplet speeds of up to 21,000 $\mu\text{m/s}$ (Ban and Nakata, *J. Phys. Chem. B*, 2015) based on this mechanism. However, such approaches have yet to create emulsion droplets that can reversibly and controllably alter their speed and directionality in response to a chemical concentration gradient as single-phase droplets of isotropic composition display selective unidirectional motion into the direction towards a minimization of interfacial energy. Droplets that can adapt to variations in their chemical environment and selectively control their speed and directionality towards or away from a chemical source have great potential utility in the context of imparting adaptive material properties and autonomous decision making within synthetically-minimal soft matter colloids.

➤ To address this apparent communication error and to highlight the advances of this work with regard to previous literature, we expanded, clarified, and improved comparisons to the literature in the revised manuscript (Čejková, J., Hanczyc, M. M., *et al.*, *Langmuir*, 2014), including alternative propulsion mechanisms of droplets to Marangoni flow (Singh, A. V., Sitti, M. *et al.*, *ACS Nano*, 2017). The respective paragraphs have been updated in the revised version of our manuscript, as follows:

- *Droplets can chemotax in response to small interfacial tension variations, driven by Marangoni-type fluid flows^{27,28,29}. Imbalances in interfacial tension across the droplet surface cause Marangoni-type fluid convections from regions of low interfacial energy to regions of high interfacial energy. These fluid flows develop along the droplet interface, which induce a commensurate convective flow inside the droplets. Small lateral variations in interfacial tension thereby suffice to achieve significant Marangoni stress-induced fluid currents at the interface that can propel a motion of the droplets³⁰. The induced currents occur in the direction of regions with higher surface tension, which drives a chemotactic locomotion of the emulsion droplets in the opposite direction. Interfacial tension gradients across a droplet surface can be evoked either chemically by variation of the surfactant effectiveness or concentration^{31,32,33,34,35}, or via fluctuations in the surfactant density at the droplet interface caused by anisotropic micellar solubilization of the dispersed phase^{36,37,38,39}. Depending on the underlying propulsion mechanism, the directionality of the overall droplet movement can be random or unidirectional^{20,36,39,40,41,42,43}, with the motion being sustained until the transient interfacial tension anisotropy equilibrates, at which point the droplets become stationary.*

An anisotropic dispersed phase solubilization into surfactant micelles can cause gradients in the surfactant density across a droplet interface, which initiate Marangoni-type convective fluid flows that drive droplet motion. In this mechanism, droplet motion can be sustained and droplets become self-propelled due to repulsion away from their own swollen micelle trail. Droplet velocities up to 400 $\mu\text{m s}^{-1}$ have been reported for this propulsion mechanism⁴⁴. In turn, for surfactant-stabilized droplets responding to an external chemical gradient, velocities up to 21,000 $\mu\text{m s}^{-1}$ have been reported^{27,45, 46}. In this mechanism, droplets act as motile sensors to interfacial tension gradients caused by local changes in the surfactant composition, strength, or effectiveness, for instance caused by chemical reactions of stimuli-responsive surfactants. Marangoni-type fluid flows from regions of low interfacial energy to regions of high interfacial energy drive the droplet locomotion. Consequently, single-phase droplets of isotropic composition

display selective unidirectional motion in the direction towards a minimization of interfacial energy and such approaches have yet to create emulsion droplets that can reversibly and controllably alter speed and direction of motion.

- Via introducing an anisotropy to the internal droplet morphology and composition, an asymmetry in the fluid flows surrounding the droplet and thus a controllable selective and dynamically tunable directionality and speed of the emulsion droplet locomotion could be achieved. With regard to the comment on an insufficient qualitative description of this active motion and a lacking flow field characterization, we have now significantly extended the respective paragraph in the revised version of the manuscript, including an updated figure and a visualization of the competitive fluid flows adjacent to the two phases of Janus emulsions inside a laminar surfactant gradient by means of PIV. The paragraph now reads:

- *To understand the differences in the directionality of droplet motion we started by comparing the interfacial tensions of both individual droplet phases inside 1 wt.% surfactant solutions of different compositions (Figure 2d). The measurements revealed a strong dependency of the interfacial tension of decane on the respective surfactant composition with significantly lowered values measured inside a pure AOT-containing aqueous phase. This difference was less pronounced for methoxyperfluorobutane, however with slightly lowered interfacial tension values inside Zonyl-containing continuous phases. These measurements illustrate that the two external interfaces of Janus droplets experience opposite interfacial tension gradients ($\nabla\gamma$) when placed inside a laminar gradient in surfactant composition inside the channel. We concluded that the resulting Marangoni-type fluid flows on the upper HC hemisphere and the bottom FC hemisphere of Janus droplets were directed into the opposite direction and thus in competition to each other. The overall droplet movement and its direction is then determined by the dominant flux of the two induced competitive fluid flows.*

To visualize the competitive fluid flows surrounding a Janus droplet exposed to a surfactant gradient, we dispersed tracer particles into the surrounding aqueous continuous phase and mapped the flow fields using particle imaging velocimetry. When subjected to a surfactant gradient, substantial fluid motion could be perceived in the vicinity of the biphasic droplets (Figure 2e). The flow profiles revealed that the direction of the Marangoni flows adjacent to each of the two different phases of a Janus droplet pointed into the opposite direction, for each phase from the region of low to regions of high interfacial tension. In these experiments, tracer particles inverted their movement direction at the triple phase contact line, which was observed to be independent of the droplet internal morphology (Figure S5). The two competitive fluid flows adjacent to each phase result in droplet displacement forces pointing in opposite directions and we concluded that, at low Reynolds number, the directionality and the speed of the overall droplet displacement overcoming the Stokes drag is dictated by the net force resulting from the two competitive Marangoni flows.

To reveal the dependency of directionality and velocity of the droplet motion solely on the power of the individual anisotropic Marangoni flows, we recorded experimentally the Janus droplet motion as a function of different surfactant compositions in the originally placed continuous phase and thus varying induced interfacial tension gradients and starting droplet morphologies (Figure S6). As the graph reveals (Figure 2g), when we added a pure AOT solution to initiate movement, droplets in a perfect

Janus morphology ($\theta = 90^\circ$; original surfactant composition: $f(\text{AOT})=0.36$) moved towards the open end of the channel, as a result of a dominant flux adjacent to the HC hemisphere. A comparison of the determined droplet velocities with previous measurements for single phase HC droplets revealed that the overall Janus droplet motion was significantly slower, which corroborated the presence of a competitive force originating from the Marangoni flow adjacent to the FC phase.

Upon expansion of the HC-W interface of droplets placed inside the channel via increasing the AOT fraction of the originally placed continuous phase prior to initiating a surfactant gradient, we observed an increase of the measured droplet velocities. This was despite the fact that the interfacial tension difference of the HC phase between the original and added surfactant solution was smaller, which would consequently result in a lowered displacement force. These experiments illustrated that the force balance between the competitive fluid flows determining the droplet chemotactic motion was not only dependent on the respective interfacial tension gradients at the two interfaces but also the ratio of exposed surface areas. Assuming a linear local interfacial tension gradient across a droplet interface, which is an appropriate approximation given that the diameter of the droplets are much smaller than the overall channel length, the net force acting on a droplet due to the competitive interfacial tension gradient-induced Marangoni flows can be expressed as

$$m \frac{dv_{\text{drop}}}{dt} = F_{\text{HC}} + F_{\text{FC}} - F_{\text{D}} \cong \int_A \frac{\delta\gamma_{\text{HC}}}{\delta x} + \int_A \frac{\delta\gamma_{\text{FC}}}{\delta x} - 6\pi r \eta v_{\text{drop}}$$

where m is the droplet mass, v_{drop} is the droplet velocity, F_{HC} and F_{FC} are the forces adjacent to the individual Janus droplet phases, F_{D} is the fluid-fluid Stokes drag force, r the droplet radius, η the viscosity of the continuous aqueous phase, and $\delta\gamma$ is the interfacial tension differential generated over the channel length x , which can thus be either positive or negative depending on the added probe⁴⁶.

Thus, the perfect Janus droplet ($\theta = 90^\circ$) with equal surface areas ($A_{\text{HC}} = A_{\text{FC}}$) moved with respect to the dominant interfacial tension gradient across the HC interface ($F_{\text{HC}} > F_{\text{FC}} + F_{\text{D}}$). With an increasing fraction of the HC surface area of the overall droplet surface area $f_{\text{HC}} = A_{\text{HC}}/(A_{\text{FC}} + A_{\text{HC}})$, the driving force for droplet displacement from fluid flows adjacent to the HC interface became more dominant, increasing droplet velocities. In turn, when the area of the fluorocarbon interface was dominant ($A_{\text{HC}} \ll A_{\text{FC}}$), the overall force originating from the Marangoni flows adjacent to the FC interface could overcome the competitive force generated by the flows along the hydrocarbon hemisphere, despite the increased interfacial tension gradient across the hydrocarbon interface ($(|\nabla\gamma_{\text{HC}}| > |\nabla\gamma_{\text{FC}}|)$). Consequently, experimental observations revealed a decrease in droplet speed upon gradual decrease of the exposed HC surface area until at a contact angle of $\theta = 30^\circ$, droplets started to move into the opposite direction (Figure 2g). Thus, both the respective interfacial tension gradients as well as the droplet morphology mutually dictated the directionality and speed of the Janus droplet motion, and Figure 2f displays a conceptual overview of the expected droplet motion resulting from a combination of the two independent factors. The physical considerations also reveal that, in theory for the case of a freely floating droplet, already very small differences in interfacial tension gradients across the two droplet interfaces ($(|\nabla\gamma_{\text{HC}}| - |\nabla\gamma_{\text{FC}}|)$) on the order of 25 mN m^{-2} , or synonymously, very small interfacial tension differentials on the order of $10^{-3} \text{ mN m}^{-1}$ across a single droplet suffice to induce a significant net fluid flow that is required to achieve droplet speeds of $v_{\text{drop}} > 100 \text{ }\mu\text{m s}^{-1}$. To further illustrate the morphology-

dependent directionality and speed of chemotactic motion, we plotted the respective droplet velocities as a function of the fraction of HC surface area with respect to the droplet surface area (Figure 2h). The resulting plot displays a linear correlation (Figure 2h), illustrating the direct proportionality between the droplet velocity and the ratio of exposed surface areas.

• I don't see the novelty concerning the usage of photoactivable surfactant, as there are many similar articles like: https://www.baigllab.com/docs/PDF/Diguet_Photomanipulation.pdf. I don't see where using an active janus is an advantage anywhere here.

➤ We appreciate this feedback from the Reviewer. Indeed, we fully agree that azobenzene-based surfactants are well known representatives of stimuli-responsive surfactants and have also been used previously in droplet movement schemes. Thus, it was not our intention to re-state this finding as a novelty and we have carefully revised our paper for potential misleading sentences in this regard. Generally, in the second part of our manuscript we move from studying a solely statically induced chemotactic motion of the Janus emulsions towards demonstrating the implications of this newly revealed chemotactic droplet behavior by initiating *in-situ* an adaptive response to light, chemicals, and bacteria, mediated via selective sensitization of Janus droplet interfaces with stimuli-responsive surfactants. Based on well-known and previously reported AzoTAB surfactants we realized, for the first time, a programmable (i.e. morphology determined) and reversible chemotactic response of emulsion droplets selectively towards or away from a UV light source.

- *I have the same comments concerning the design and control of the flow motion of active Janus droplet with similar perfluorated surfactant: as a much more detailed article was published online recently, including a much more detailed flow control description and chemical description (Meredith, C., et al. Chemical design of self-propelled Janus droplets. Preprint at <https://doi.org/10.33774/chemrxiv-2021-p7jvq-v2> (2021). REF37 in the manuscript).*

- We thank the reviewer for this comment. As mentioned above in response to the very first comment, we have taken a number of steps to address an apparent communication error within the originally submitted version of the manuscript. Alongside an extended physical discussion of different propulsion mechanism of emulsion droplets aimed to present a more-clear, and accurate depiction of the current state-of-the-art of research on motile emulsion droplets, we have, in particular, sought to address and emphasize the differences between two very different underlying mechanisms for a chemotactic droplet motion: a movement initiated via gradients in surfactant composition or effectiveness as studied in our paper, as opposed to a motion initiated and driven by micellar solubilization of the dispersed phase, as it is present in the excellent work by Meredith, C., et al.

- *Also, the chemical composition used here, prevents any kind of bio applications of these Janus active droplets, which is another strong limitation.*

- We thank the reviewer for this comment and, of course, we agree that the non-aqueous dispersed emulsion phases are not biocompatible. However, with regard to biological applications of the Janus droplets used in our study we would like to point out that such systems have recently been explored and established as powerful transduction and signal amplification elements in biosensing applications. Depending on the underlying interaction, i.e. the immobilized cellular recognition motif at the droplet interfaces, multivalent binding interactions with a biological target can result in droplet agglutination, or competitive binding interactions with reversibly sensitized droplet interfaces can transduce such interactions into morphological changes. In these sensing paradigms, microscale changes in droplet morphology or alignment lead to macroscopic changes in their optical properties which allows the generation of facile optical read-out sensing paradigms. Simply put, biological colloids such as bacteria or cells, can interact with the interfaces of droplets in many ways similarly to how cells interact with each other. These works demonstrate that droplets can respond to biological stimulation, which was essential for the realization of a reversibly directed motion and sensitivity of our Janus droplets in response to the presence of bacteria, as presented in the last part of our manuscript. Such systems could form the basis for the development of novel sensing platforms based on motile and adaptive sensing elements. In addition, the experimental realization of motile emulsion droplets with programmable speed and directionality based on organic fluids in this paper has implications also for droplets of different composition, including aqueous droplets and we have added a literature reference for such a microswimmer example that allows for biological compatibility (Holler, S., Hanczyc, M. M., *Sci. Rep.*, 2020).

- The article redaction is moreover confusing as many infos are put in suppmat but discussed in detail into the main text, which made this article not suitable for a broad audience (see the mention to figure S3 page 6.)

➤ We thank the reviewer for this comment and appreciate the suggestion for increasing the legibility of our manuscript for a broad audience. In addition to changes we made with regard to including more data in the manuscript in response to some other comments, we have also updated Figure 1 in the revised version of the manuscript. It now includes an additional scheme and the data for the chemotactic motion of single phase isotropic droplets that are discussed in the first part of our manuscript. To support the respective written paragraphs in the manuscript, this Figure now displays the data for the chemotactic motion of the droplets initiated by variations of the surfactant composition within the continuous phase.

- Also, I have a big doubt concerning the reality of the results concerning the measured active Janus droplet velocities present in supp video 1 and 2. These videos seem to show a laminar driven flow BEHIND and not a purely active flow. In particular, an extremely similar other system in chemical composition (the one in ref37) presents Janus droplets velocities that were 1 order slower (even nearly 2 orders), which looks a confirmation to me that the authors did not performed the experiments correctly and measured the motion of active droplets into a strong driven flow.

➤ We thank the Reviewer for this concern, however can state with full confidence that the reported droplet motion is purely active flow and measured velocities are real: Within the reported experimental setup a surfactant gradient was evoked by addition of a concentrated surfactant solution to the inlet of the channel. The channel was sealed to circumvent evaporative volume changes and any capillary flows. In addition, after the evocation of the surfactant gradient there was a time lapse of a few seconds before droplet acceleration started due to diffusion of the added surfactant solution into the channel. We can therefore exclude any flow from behind. Moreover, as we hope was conveyed in our answer to the first comment

above, the measured droplet velocities are absolutely within the range of previous reports for a chemotactic droplet motion that is based on interfacial tension gradients. Motion based on this mechanism is generally at least one order of magnitude faster as compared to droplets moving in response to a temporal or sustained anisotropy in surfactant density across a droplet surface due to micellar solubilization of the dispersed phase, which constitutes a very different alternative propulsion mechanism, a.o. studied in ref 37 (Meredith et al, Matter, 2022).

- *Behind these criticisms, I only detect two interesting points. One the droplet active motion is extremely fast, however the reason is not explained by the authors. I believe these results came from experiments that were not performed correctly. Also, the part using the enzymatic reaction is interesting but also not really novel... for an unclear reason, it is even barely discussed by the authors.*

- We thank the reviewer for this comment. As outlined above, the underlying mechanism supports the rapid response of the microdroplets to variations in surfactant composition of effectiveness, and the latter can be used to direct *in-situ* their adaptive response to light, chemicals, and bacteria, mediated via selective sensitization of Janus droplet interfaces with stimuli-responsive surfactants. In fact, our physical considerations revealed that, depending on the trigger and the internal droplet composition and geometry droplet speeds can be very fast, fine-tuned, and altered. Very small gradients in interfacial tension $\nabla\gamma$ across the droplet interface suffice to achieve overall droplet speeds of several hundred $\mu\text{m/s}$. Taking advantage of this sensitive response to variations in the surfactant effectiveness, we realized a programmable chemotactic response of Janus emulsions to the presence of bacteria. The transduction mechanism was based on a sensitization of the hydrocarbon water interface with enzyme cleavable surfactants. We have investigated a reduction of the hydrocarbon-water interfacial tension as a result of surfactant cleavage by the bacterial exoenzymes and have updated the last figure of our manuscript to better reflect these experiments in the manuscript.

Overall, we would like to thank the reviewer again for the detailed and critical feedback. We hope that we are addressing all remaining concerns and could convince the reviewer particularly with regard to the reality of the reported results. We are convinced the feedback helped to improve the presentation of our findings in the revised version of our manuscript.

Reviewer #2: *Authors report the chemotactic actuation and motion of Janus emulsion droplets that respond to surfactant gradients. The work is original, and the story is scientifically convincing. I am impressed by their strong efforts, e.g., their thorough datasets in SI and the experiments with light, chemicals, and bacteria, to demonstrate the implications of the chemotactic behavior of the Janus emulsion droplets. I believe this work would be of interest to the soft matter community. I want to recommend the publication of the manuscript in Nature Communication, only after improving some drawbacks.*

- We are grateful for the positive feedback and thank the reviewer for their time to review our manuscript and for the many helpful and constructive suggestions to improve the presentation of our findings.
- *I have had difficulties understanding some details of the work. It is partly because of their mistakes in manuscript writing. To name a few, the caption does not match the plots in Figure 2 and Figure 5. Line 387 says the wavelength is 460 nm but 470 nm in Figure 5. I am also afraid that some choices of wordings are rather clumsy. For example, in line 395, I presume "continuously" is better than "permanently." I have spotted similar phrases/sentences/mistakes throughout the manuscript.*
 - We thank the reviewer for this comment and the detailed suggestions and edits on our manuscript. We have updated the mentioned captions for Figure 2 and Figure 5 to properly reflect data. For mentions of the blue light wavelength the manuscript has been corrected accordingly and we have updated the language on line 395 per the Reviewers suggestion. To avoid further communication errors and ensure accurate scientific description of our results we have carefully revised the whole manuscript and the corresponding sections are highlighted in the submitted revised version of the manuscript.
 - *The authors highlighted the "intrinsic out-of-eq" nature of emulsion droplets in the abstract. I agree, as written in line 60, the emulsion droplets are at a meta-stable state. I want to ask the authors what "out-of-eq" aspects of the emulsion droplets make them a suitable model system. For instance, the exchange of the surfactant molecules between the interface and continuous phase is just a dynamic 'equilibrium,' not "out-of-eq," isn't it? Please enlighten readers and me.*
 - We very much appreciate this comment by the reviewer and carefully revised the respective sentences of the manuscript in this regard. Indeed, our microemulsion droplets represent a metastable system. Key to the realization of our results is the multiresponsiveness of the droplets with regard to the reconfigurability of the internal droplet geometry and the chemotactic responses. These readily observable effects in response to marginal variations in interfacial tension are based on the ability for a stimuli-triggered evocation of an out-of-equilibrium state that then drives droplet motion and morphological reconfiguration.
 - *Because the gravitational alignment is not mentioned in the abstract, the upper and lower hemisphere (line 24) sound awkward.*
 - We thank the reviewer for this comment. We updated the abstract to aid the reader's understanding of the system by referring to either droplet 'phase' instead of hemisphere, and have updated the main manuscript text accordingly.

- *I wonder what the authors mean by "permanently asymmetric" in line 88. Isn't Janus colloid (e.g., catalyst-coated) permanently asymmetric?*

- Thank you for this comment. In the respective paragraph of the manuscript we were attempting to highlight the difference between solid particles and droplets, and more specifically the static nature of a solid particle's morphology and behavior, when compared to a dynamic, reconfigurable droplet. To address this, we updated the manuscript (previously on line 88) to differentiate particles from droplets, which now states:

"Whereas solid Janus colloids with two different chemistries on each side of the particle have found widespread application in the design of active microswimmers with directional propulsion profiles, an asymmetric, dynamic, fluid-based system such as spherical biphasic Janus droplets has not been explored previously in this context."
 (...of droplet motion driven by chemically induced surfactant gradients)

- *I believe the authors should be cautious when they mention the interfacial tension gradient. They have characterized their system by interfacial tension 'differential' (difference in the interfacial tensions at two ends), not by the gradient, which is the difference divided by the channel length (20 mm according to the SI), i.e., the slope, if the tension changes linearly. What determines the speed is not the difference but the gradient. Specifying the gradient would be essential to compare the authors' data with other experiments. Furthermore, $f(AOT)$ in plots is a handy parameter for the experiments but does not give direct information about the difference and gradient; I see Figure S3 gives some details.*

- We thank the reviewer for this comment and insight. We have updated Figure 1 in order to clarify the message of single phase droplet chemotaxis induced by gradients in surfactant composition, where the total surfactant concentration inside the continuous phase remains constant (1 wt.%). We have also carefully revised the manuscript text with regard to the motion of droplets within gradients in surfactant composition. With regard to the chemotactic motion of the Janus emulsion droplets in response to such gradients in interfacial tension across the two droplet interfaces we have updated the respective paragraph on the qualitative description of our system, which now reads:

- *These measurements illustrate that the two external interfaces of Janus droplets experience opposite interfacial tension gradients ($\nabla\gamma$) when placed inside a laminar gradient in surfactant composition inside the channel. We concluded that the resulting Marangoni-type fluid flows on the upper HC hemisphere and the bottom FC hemisphere of Janus droplets were directed into the opposite direction and thus in competition to each other. The overall droplet movement and its direction is then determined by the dominant flux of the two induced competitive fluid flows.*

To visualize the competitive fluid flows surrounding a Janus droplet exposed to a surfactant gradient, we dispersed tracer particles into the surrounding aqueous continuous phase and mapped the flow fields using particle imaging velocimetry. When subjected to a surfactant gradient, substantial fluid motion could be perceived in the vicinity of the biphasic droplets (Figure 2e). The flow profiles revealed that the direction of the Marangoni flows adjacent to each of the two different phases of a Janus droplet

pointed into the opposite direction, for each phase from the region of low to regions of high interfacial tension. In these experiments, tracer particles inverted their movement direction at the triple phase contact line, which was observed to be independent of the droplet internal morphology (Figure S5). The two competitive fluid flows adjacent to each phase result in droplet displacement forces pointing in opposite directions and we concluded that, at low Reynolds number, the directionality and the speed of the overall droplet displacement overcoming the Stokes drag is dictated by the net force resulting from the two competitive Marangoni flows.

To reveal the dependency of directionality and velocity of the droplet motion solely on the power of the individual anisotropic Marangoni flows, we recorded experimentally the Janus droplet motion as a function of different surfactant compositions in the originally placed continuous phase and thus varying induced interfacial tension gradients and starting droplet morphologies (Figure S6). As the graph reveals (Figure 2g), when we added a pure AOT solution to initiate movement, droplets in a perfect Janus morphology ($\theta = 90^\circ$; original surfactant composition: $f(\text{AOT})=0.36$) moved towards the open end of the channel, as a result of a dominant flux adjacent to the HC hemisphere. A comparison of the determined droplet velocities with previous measurements for single phase HC droplets revealed that the overall Janus droplet motion was significantly slower, which corroborated the presence of a competitive force originating from the Marangoni flow adjacent to the FC phase.

Upon expansion of the HC-W interface of droplets placed inside the channel via increasing the AOT fraction of the originally placed continuous phase prior to initiating a surfactant gradient, we observed an increase of the measured droplet velocities. This was despite the fact that the interfacial tension difference of the HC phase between the original and added surfactant solution was smaller, which would consequently result in a lowered displacement force. These experiments illustrated that the force balance between the competitive fluid flows determining the droplet chemotactic motion was not only dependent on the respective interfacial tension gradients at the two interfaces but also the ratio of exposed surface areas. Assuming a linear local interfacial tension gradient across a droplet interface, which is an appropriate approximation given that the diameter of the droplets are much smaller than the overall channel length, the net force acting on a droplet due to the competitive interfacial tension gradient-induced Marangoni flows can be expressed as

$$m \frac{dv_{\text{drop}}}{dt} = F_{\text{HC}} + F_{\text{FC}} - F_{\text{D}} \cong \int_A \frac{\delta\gamma_{\text{HC}}}{\delta x} + \int_A \frac{\delta\gamma_{\text{FC}}}{\delta x} - 6\pi r \eta v_{\text{drop}}$$

where m is the droplet mass, v_{drop} is the droplet velocity, F_{HC} and F_{FC} are the forces adjacent to the individual Janus droplet phases, F_{D} is the fluid-fluid Stokes drag force, r the droplet radius, η the viscosity of the continuous aqueous phase, and $\delta\gamma$ is the interfacial tension differential generated over the channel length x , which can thus be either positive or negative depending on the added probe⁴⁶.

- After dropping the surfactant solution to the one end of the channel, how long does it for the channel to attain the linear γ gradient? How can we tell?
- We thank the reviewer for this question. Experimentally, movement would occur on the order of seconds after addition of the surfactant, attributed to diffusion of surfactant into the channel. To address this comment, we conducted an experiment inside the channel where Janus droplets were first placed as densely packed monolayer, and subsequently, a 1 wt.% Zonyl solution was

applied to the channel inlet. We then characterized changes in droplet morphology to measure the progression of the surfactant diffusion. We have added a figure of these results as well as the respective video to the revised version of the supporting information. The surfactant diffusion progressed at a rate of ~ 0.11 mm/s.

- “b) Optical micrographs of Janus droplets dispersed inside a 1 wt.% surfactant solution of 4:6 SDS:Zonyl, placed as a densely packed monolayer inside the channel. Upon addition a 1 wt.% Zonyl solution to the inlet of the channel, the progression of the surfactant diffusion can be observed via droplet morphology changes throughout the channel. Dotted lines represent the progression of surfactant diffusion at $t = 0s$, and $t = 30s$ respectively.”

- The interfacial tension values from the tensiometry may need more info about its accuracy/precision. They are from the fitting of the interface, and the authors claim the roles of its subtle difference in line 186.

- We thank the reviewer for raising this concern. To address this we have updated the methods section of the revised manuscript version to provide a more detailed description of the performed measurements:

- Pendant drop tensiometry was performed on a Krüss DSA10-MK2 drop shape analyzer. Time-resolved interfacial tension measurements were taken from needle-attached droplets that were imaged, fit, and analyzed by Krüss Advance software via Laplace equation. Final interfacial tension values were calculated via fitting time resolved measurements recorded until the exponential decrease of interfacial tension due to surfactant adsorption lowered at a rate of $< 0.1 \text{ mN m}^{-1} \text{ min}^{-1}$ ⁴⁶. Measurements were performed between constituent dispersed-phase oils and binary surfactant solutions utilized to generate droplets of particular morphologies, or at points along the binary surfactant gradient from $f(\text{AOT}) = 0$ to 1 versus Zonyl in 1 WT% solutions, or $f(\text{SDS}) = 0$ to 1 versus Zonyl in 1 WT% solutions. Needles and cuvettes were cleaned with water, acetone, and dried between measurements.

- I have trouble understanding how and why MPFB droplets move away from the region of high HC surfactant concentration. The authors give the interfacial tensions inside pure AOT and Zonyl and argue they have a small difference. Specifying the situation, i.e., the initial background type/concentration of surfactant and the tension gradient, may help.

- We thank the Reviewer for another very helpful feedback. We have updated the text to specify the surfactant composition inside the originally placed continuous phase. We have also updated Figure 1 to display and highlight the single-phase droplet movement in response to an evoked

gradient in surfactant composition at constant overall surfactant concentration, as opposed to a previously reported motion in response to gradients in surfactant concentration. In these experiments, the overall surfactant concentration inside the channel was held constant (1 wt.%). The pure single-phase MPFD droplets displayed an increased interfacial tension inside surfactant solutions with an increased fraction of HC surfactant. Consequently, the addition of a pure AOT solution to the inlet of the channel resulted in chemorepulsion and FC droplets moved away.

- *Showing the axes (x,y,z) or the direction of gravity will be helpful. For instance, in Figure 2 b&c. In the same vein, the side-view schematic seems quite different from the experimental side-view image, which is almost in an engulfed configuration. Can the authors comment on this? Additionally, is the authors' Marangoni convection scheme still valid when they are in the engulfed configuration?*
- We thank the Reviewer for this comment. We have carefully revised every figure in this regard and added directional notation (XY, ZX) to aid the reader in distinguishing between side-view and vertical images of the droplets. The morphology of droplets is dictated by the calibration curves provided in the supporting information. Prior to all experiments droplet morphologies were recorded in side-view or the internal morphology of droplets was specifically adjusted based on this calibration curve. We can state with full certainty that the droplets in Figure 2 b and c are in a Janus configuration and side-view micrographs imaged through the channel may not always reflect well the ‘opened-up’ morphology of the droplets. Double emulsion droplets in a fully engulfed configuration respond to the chemical gradient similar to single phase, isotropic droplets unless variations in surfactant composition result in morphological changes. As the presented droplets are both Janus, interfaces are responding to the opposed interfacial tension gradients, and the schemes are presented for clarity to the reader.

- *I recommend the authors should think again about the meaning of Figure 2e. With the fixed volume ratio of the two phases, the surface area ratio is geometrically related to the contact angle; they can be calculated analytically with no help from experiments.*
 - We thank the Reviewer for their comment, and agree that the content within the Figure 2e of the original manuscript was unnecessary. Our aim was to illustrate the connection between droplet morphology and surface area ratio for a general audience. We have now moved the figure from the main text to the revised supporting information (figure S6).
- *To control the starting droplet morphologies, the authors dispersed the droplets into the continuous phases of different surfactant concentrations. These different initial concentrations should affect the gradient when the authors add the surfactant solution to the other end of the channel. Can the authors comment on this?*
 - We thank the reviewer for this comment. Indeed, upon initiation of the chemotactic droplet motion by adding a pure surfactant solution, e.g. 1 wt.% AOT, different surfactant gradients are generated inside the channels as a result of the different compositions of the originally placed continuous phases. To clarify and better describe the mutual dependency of the resulting direction and velocity of chemotactic droplet motion on both the respective interfacial tension gradients across the two different interfaces as well as the droplet morphology, we have added a conceptual overview of the expected droplet motion resulting from a combination of the two independent factors and updated the respective paragraphs of our manuscript:
 - *... the perfect Janus droplet ($\theta = 90^\circ$) with equal surface areas ($A_{HC} = A_{FC}$) moved with respect to the dominant interfacial tension gradient across the HC interface ($F_{HC} > F_{FC} + F_D$). With an increasing fraction of the HC surface area of the overall droplet surface area $f_{HC} = A_{HC} / (A_{FC} + A_{HC})$, the driving force for droplet displacement from fluid flows adjacent to the HC interface became more dominant, increasing droplet velocities. In turn, when the area of the fluorocarbon interface was dominant ($A_{HC} \ll A_{FC}$), the overall force originating from the Marangoni flows adjacent to the FC interface could overcome the competitive force generated by the flows along the hydrocarbon hemisphere, despite the increased interfacial tension gradient across the hydrocarbon interface ($(|\nabla\gamma_{HC}| > |\nabla\gamma_{FC}|)$). Consequently, experimental observations revealed a decrease in droplet speed upon gradual decrease of the exposed HC surface area until at a contact angle of $\theta = 30^\circ$, droplets started to move into the opposite direction (Figure 2g). Thus, both the respective interfacial tension gradients as well as the droplet morphology mutually dictated the directionality and speed of the Janus droplet motion, and Figure 2f displays a conceptual overview of the expected droplet motion resulting from a combination of the two independent factors. The physical considerations also reveal that, in theory for the case of a freely floating droplet, already very small differences in interfacial tension gradients across the two droplet interfaces ($(|\nabla\gamma_{HC}| - |\nabla\gamma_{FC}|)$) on the order of 25 mN m^{-2} , or synonymously, very small interfacial tension differentials on the order of $10^{-3} \text{ mN m}^{-1}$ across a single droplet suffice to induce a significant net fluid flow that is required to achieve droplet speeds of $v_{drop} > 100 \mu\text{m s}^{-1}$. To further illustrate the morphology-dependent directionality and speed of chemotactic motion, we plotted the respective droplet velocities as a function of the fraction of HC surface area with respect to the droplet surface area*

(Figure 2h). The resulting plot displays a linear correlation (Figure 2h), illustrating the direct proportionality between the droplet velocity and the ratio of exposed surface areas.

- The authors should be more quantitative about the time scale of the morphology change, instead of "much slower" in line 281. Does the droplet maintain the same morphology (and the resulting same speed) when traveling along the gradient? In a similar vein, f Area in Fig. 3c is the area fraction in the starting morphology?

➤ We thank the Reviewer for this comment. To quantify the time correlation we conducted an additional experiment, in which we determined a time delay of 22 sec between the initial onset of chemotactic movement and observable morphological changes. It should be noted that droplets travelling down the channel, in principle, experience a different interfacial tension gradient and morphological rearrangements not necessarily result in a decay of the motion (except for the boundary conditions where the forces originating from the competitive fluid flows along the upper and lower hemisphere are equal). In our experiment, we monitored droplet movement in response to an addition of AOT (1 wt.%) to the inlet of the channel. Inside the monitored section of the channel, we observed, in parallel to a motion of the vast majority of droplets, a single droplet that was restrained from moving due to a defect in the channel. At hand of this droplet, we recorded the morphological transition by means of quantifying the contact angle over time. The resulting plot has been added to the supporting information (Figure S12).

- “a) Measurement of the time delay for the onset of a morphological transition of Janus droplets in response to changes in surfactant composition with respect to the instantaneous onset of chemotactic motion as a result of an evoked interfacial tension gradient; $t = 0$ marks the time at which freely moving Janus droplets began to move in response to an evoked surfactant gradient inside the channel. Inside the monitored section of the channel, we observed, in parallel to droplet motion, a droplet that was restrained from moving due to a defect in the channel and recorded the contact angle of this droplet at the same position over time.

- Fig. 4c needs more explanation. What configuration is changing into what configuration? In the figure, what is what? I cannot discriminate the tilted one from the reconfigured one.

- We thank the reviewer for this comment. We have updated the respective figure in the revised version of the manuscript where we added a schematic depiction of the droplet arrangements inside the surfactant gradient.

- What is the meaning of "a mechanically confined" in line 405?

- We thank the Reviewer for pointing this out. ‘Mechanically confined’ was referring to the inability of the droplet to move, as they were originally placed as a densely packed monolayer. We have updated the respective text paragraph accordingly: “To this end, we mapped the flow profiles surrounding a single droplet within a densely packed monolayer in the vicinity of the UV light beam, which was therefore restrained from moving”.

Reviewer #3: *This paper describes the migration of biphasic Janus droplets in surfactant gradients. Homogeneous (non-Janus) drops move to regions of lower interfacial tension (high surfactant concentration) via Marangoni flows at the drop interface. The Authors demonstrate how drops of different liquids can move in different directions when position in a common gradient containing two surfactants. This observation provides a basis for creating spherical Janus droplets that can move up or down surfactant gradients depending on their internal morphology. The Authors show how the Janus morphology can change over time thereby reversing drop motion in the gradient. In addition to drop migration, the Authors quantify the gradient-induced rotation of Janus droplets from their preferred gravitational alignment. The Authors explore other routes for generating gradient-induced motions using reactive surfactants based on photoisomerization and enzymatic cleavage.*

The use of Janus morphology to tune the gradient-induced migration of droplets within environmental gradients is an interesting new capability. The fact that this internal degree of freedom is also responsive to the environment has implications for designing autonomous behaviors based on internal feedback mechanisms (see, for example, Alvarez & Isa, Nat Commun, 2021, doi:10.1038/s41467-021-25108-2). The paper is very clearly written, and the conclusions are well supported by high quality experiments. In hindsight, the results are not particularly surprising; however, the experimental realization of this concept is non-trivial and merits publication in Nature Communications.

- We thank the Reviewer for their time reviewing our manuscript, and share in their optimism for the implications of morphologically-determined response to interfacial tension gradients. We are especially grateful that the Reviewer found our insights suitable for publication in *Nature Communications*. We appreciate the reviewers suggestion for the reference with regard to the design of adaptive multi-responsive colloids, which we have added to the revised version of the manuscript.

The Authors should cite related work on Janus droplets by the Zarzar group (Chemical Design of Self-Propelled Janus Droplets, doi: 10.26434/chemrxiv.14378780.v1).

- We thank the Reviewer for their suggestion. The specific reference has now been published in the journal *Matter*, and is cited our manuscript.

REVIEWERS' COMMENTS

Reviewer #1 (Remarks to the Author):

The authors have succeed to change my opinion on this manuscript. They include piv analysis which remove my concern about the potential existence of an underlying applied driving passive flow in superposition of the active flow propulsion. The novelty of the work is now clearly describe in relation with the current literature. Thus, I recommend this paper for publication and I congrat the authors for their efforts while considering my concerns/doubts.

Reviewer #2 (Remarks to the Author):

I find the authors thoroughly address my comments/questions in the revised manuscript, with additional experiments, renewed figures, and detailed descriptions. I recommend the publication of the revised manuscript in Nature Communications. I provide just a couple of minor comments, which may improve the work before the final publication.

1. The authors' arguments are based on the assumption of a LINEAR surfactant (or interfacial tension, equivalently) gradient across the channel. But, unless there are constant source and sink in this 1D system, the concentration profile would be different from the linear one. I believe, The authors' system with the surfactant solution dropping instead corresponds to the diffusion of instantaneous source, i.e., a gaussian that spreads as time goes by. Can authors defend their assumption of the linear profile? I do not think this qualitatively affects the conclusion; for example, a linear gradient across the droplet would still be good enough. But, could the authors comment on possible amendments/usage because of non-linear gradient profiles?

2. In the reviewer response, the authors once mentioned "microemulsion." Some literature misuse "microemulsion" because microemulsion is a thermodynamical equilibrium phase, distinct from "small" emulsion droplets. Googling or even Wikipedia can give us a quick answer. Anyway, this comment is not of any importance in this work because microemulsion does not appear in the main manuscript.

Reviewer #3 (Remarks to the Author):

The Authors have addressed my previous comments. More importantly, they have made significant additions and improvements to the manuscript in response to other Reviewer criticisms. I support publication of the revised manuscript in Nature Communications.

Point-by-point response to concerns of Reviewer #2.

- The authors' arguments are based on the assumption of a LINEAR surfactant (or interfacial tension, equivalently) gradient across the channel. But, unless there are constant source and sink in this 1D system, the concentration profile would be different from the linear one. I believe, the authors' system with the surfactant solution dropping instead corresponds to the diffusion of instantaneous source, i.e., a gaussian that spreads as time goes by. Can authors defend their assumption of the linear profile? I do not think this qualitatively affects the conclusion; for example, a linear gradient across the droplet would still be good enough. But, could the authors comment on possible amendments/usage because of non-linear gradient profiles?
 - We thank reviewer 2 again for their time and consideration with regard to the correction of this manuscript. We agree that the concentration profile will be non-linear in experiments, where a surfactant gradient was externally evoked by dropping a solution to one end of the channel. As stated in the manuscript, we expect surfactants to progress through the channel by diffusion and thus, the interfacial tension variations will be non-linear. This may be different in the case of experiments, where the droplet motility is evoked inside channels by using light- and biochemically evoked surfactant gradients as in these experiments surfactant effectiveness is constantly impacted. In our amendment experiments highlighting the usage of the described programmable response of Janus emulsions to an oncoming surfactant gradient, we used light- and biochemically evoked surfactant gradients to evoke a droplet chemotaxis. When triggering a local light-induced AzoTAB isomerization or a continuous exoenzyme-mediated surfactant cleavage by extracellular enzymes, we expect a continuous gradient in surfactant composition throughout the channel. However, besides any non-linear diffusion profile, the interfacial tension gradient would also be affected by non-linear changes in interfacial tension values between the individual oil phases and the aqueous continuous phase containing the different ratios of surfactants. We have thus carefully double-checked and revised our manuscript for any misleading statements in this regard. In our physical consideration accompanying the discussion on the reversible and directional chemotactic motility of Janus droplets, we assume a linear local interfacial tension gradient only across a single droplet interface, which is an appropriate approximation given that the diameter of the droplets are much smaller than the overall channel length. We expect that for the particular scenario of a non-linear gradient profile, a combination of the chemotactic actuation and motile characteristics with the morphological reconfigurability of the Janus droplets may pave the path towards the design of adaptive motile droplet sensors.
- In the reviewer response, the authors once mentioned "microemulsion." Some literature misuse "microemulsion" because microemulsion is a thermodynamical equilibrium phase, distinct from "small" emulsion droplets. Googling or even Wikipedia can give us a quick answer. Anyway, this comment is not of any importance in this work because microemulsion does not appear in the main manuscript.
 - We thank the reviewer for their keen eye. Indeed, we absolutely agree with the reviewer and the correct term is 'macroemulsion', an emulsion comprised of microscale emulsion droplets.